# Competitive coordination of the dual roles of the Hedgehog co-receptor in homophilic adhesion and signal reception

**Shu Yang[1,2†], Ya Zhang[1,2†], Chuxuan Yang[3†], Xuefeng Wu[1,2], Sarah Maria El Oud[3], Rongfang Chen[1,2], Xudong Cai[1,2], Xufeng S Wu[4], Ganhui Lan[3\*], Xiaoyan Zheng[1,2\*]**

[1]Department of Anatomy and Cell Biology, George Washington University School of Medicine and Health Sciences, Washington, United States; [2]GW Cancer Center, George Washington University School of Medicine and Health Sciences, Washington, United States; [3]Department of Physics, George Washington University, Washington, United States; [4]Laboratory of Cell Biology, National Heart, Lung, and Blood Institute, National Institutes of Health, Bethesda, United States

**Abstract** Hedgehog (Hh) signaling patterns embryonic tissues and contributes to homeostasis in adults. In *Drosophila*, Hh transport and signaling are thought to occur along a specialized class of actin-rich filopodia, termed cytonemes. Here, we report that Interference hedgehog (Ihog) not only forms a Hh receptor complex with Patched to mediate intracellular signaling, but Ihog also engages in *trans*-homophilic binding leading to cytoneme stabilization in a manner independent of its role as the Hh receptor. Both functions of Ihog (*trans*-homophilic binding for cytoneme stabilization and Hh binding for ligand sensing) involve a heparin-binding site on the first fibronectin repeat of the extracellular domain. Thus, the Ihog-Ihog interaction and the Hh-Ihog interaction cannot occur simultaneously for a single Ihog molecule. By combining experimental data and mathematical modeling, we determined that Hh-Ihog heterophilic interaction dominates and Hh can disrupt and displace Ihog molecules involved in *trans*-homophilic binding. Consequently, we proposed that the weaker Ihog-Ihog *trans* interaction promotes and stabilizes direct membrane contacts along cytonemes and that, as the cytoneme encounters secreted Hh ligands, the ligands trigger release of Ihog from *trans* Ihog-Ihog complex enabling transport or internalization of the Hh ligand-Ihog-Patched -receptor complex. Thus, the seemingly incompatible functions of Ihog in homophilic adhesion and ligand binding cooperate to assist Hh transport and reception along the cytonemes.

**\*For correspondence:**
ganhuilan@gmail.com (GL);
xzheng@gwu.edu (XZ)

[†]These authors contributed equally to this work

**Competing interests:** The authors declare that no competing interests exist.

## Introduction

Hedgehog (Hh) signaling plays essential roles in patterning of multicellular embryos and maintaining adult organ homeostasis. Aberration in the precise temporal-spatial regulation and transduction of the Hh signaling pathway is involved in several birth defects (*Muenke and Beachy, 2000*) and various proliferative disorders, such as the growth of malignant tumors (*Varjosalo and Taipale, 2008*).

Hh protein precursor undergoes autoprocessing and lipid modification that generates the mature Hh ligand as an amino-terminal signaling peptide (HhN) dually modified by palmitoyl and cholesteryl adducts (*Mann and Beachy, 2004*). Intracellular signaling is triggered by binding of the dually lipidated Hh ligand to the receptor. The Hh receptor suppresses the essential downstream pathway component Smoothened (Smo) and limits the range of signaling by sequestering the Hh ligands. The Hh receptor is comprised of Patched (Ptc) and a member of the Ihog family, which in *Drosophila* is one of the functionally interchangeable proteins encoded by *interference Hedgehog* (*ihog*) or

*brother of ihog* (*boi*) (*Lum et al., 2003*; *McLellan et al., 2006*; *Yao et al., 2006*; *Camp et al., 2010*; *Chou et al., 2010*; *Hartman et al., 2010*; *Yan et al., 2010*; *Zheng et al., 2010*).

The Ihog family proteins are type I single-span transmembrane proteins with immunoglobulin (Ig) and fibronectin type III (FNIII) domains, resembling typical cell adhesion molecules in the Ig-CAM (Ig cell adhesion molecule) superfamily. Previous biochemical and structural studies showed that the first FNIII domain (Fn1) in the extracellular portion of Ihog is involved in binding to the Hh ligand (*McLellan et al., 2006*; *Yao et al., 2006*), whereas the second FNIII domain (Fn2) of Ihog interacts with Ptc. Both Fn1 and Fn2 domains are required for Hh signal reception through the formation of a high-affinity multimolecular complex of Ihog, Ptc, and Hh (*Zheng et al., 2010*). Ihog proteins not only play an essential role in Hh signal transduction but also mediate cell-cell interactions in a homophilic, calcium-independent manner (*Zheng et al., 2010*; *Hsia et al., 2017*; *Wu et al., 2019*). The region that mediates the *trans* Ihog-Ihog interaction overlaps with the region that mediates the interaction with Hh on the Ihog Fn1 domain and includes a region where the negatively charged glycan heparin binds (*McLellan et al., 2006*; *Wu et al., 2019*). Heparin is required for not only Hh-Ihog interactions but also Ihog-Ihog homophilic *trans* interactions *in vitro* (*McLellan et al., 2006*; *Zhang et al., 2007*; *Wu et al., 2019*).

The presence of dual functions as an adhesion protein and as a signaling protein is not unique to Ihog proteins. Other members of the Ig-CAM family, such as the netrin receptor DCC, the Slit receptor Robo, and neural cell adhesion molecule (N-CAM), have dual roles. These proteins act as molecular 'glue' that holds cells together and as molecular sensors to mediate cellular responses, such as motility, proliferation, and survival (*Juliano, 2002*; *Orian-Rousseau and Ponta, 2008*). However, ligand binding and cell adhesion are often structurally separated and involve different extracellular domains (*Frei et al., 1992*; *Martin-Bermudo and Brown, 1999*; *Sjöstrand et al., 2007*). In contrast, the Ihog protein couples these distinct functions within the same region. The physiological consequences of coupling two distinct functions into the same region of the Ihog protein are unknown.

In the *Drosophila* wing imaginal discs, Hh is secreted in the posterior (P) compartment and spreads toward the anterior (A) compartment (*Basler and Struhl, 1994*; *Capdevila et al., 1994*; *Tabata and Kornberg, 1994*). Hh signaling does not occur in P compartment cells because they do not express critical components of the Hh pathway, such as the major transcriptional effector Ci (*Eaton and Kornberg, 1990*). In contrast, A compartment cells can receive and respond to Hh but are unable to produce Hh. In A compartment cells located close to the source of Hh ligand production at the A/P boundary, Hh signaling triggers pathway activity and, consequently, an increase in the transcription of target genes (*Ingham et al., 1991*; *Basler and Struhl, 1994*; *Capdevila et al., 1994*; *Tabata and Kornberg, 1994*; *Chen and Struhl, 1996*). A model of Hh secretion and transport from the basal surface of the *Drosophila* wing imaginal discs epithelia involves movement of Hh along cytonemes (*Gradilla et al., 2014*; *Chen et al., 2017*; *González-Méndez et al., 2017*), which are dynamic thin cellular protrusions specialized for the intercellular exchange of signaling proteins (*Ramírez-Weber and Kornberg, 1999*; *Roy et al., 2011*; *Gradilla and Guerrero, 2013*; *Kornberg, 2014*). Intriguingly, when Ihog is co-expressed with the cytoskeletal and membrane markers of these structures, these thin cellular protrusions are much easier to be detected microscopically (*Callejo et al., 2011*; *Bilioni et al., 2013*; *Bischoff et al., 2013*; *González-Méndez et al., 2017*), which suggests that Ihog has roles in generating or stabilizing cytonemes. Moreover, overexpressed Ihog is used as a cytoneme marker to visualize these structures (*Portela et al., 2019*; *González-Méndez et al., 2020*). Yet whether and how Ihog promotes cytoneme growth or stabilization and how cytonemes contribute to Hh transport and signal reception remain poorly understood.

Here, we report that cytoneme-localized Ihog proteins engage in *trans*-homophilic binding leading to cytoneme stabilization in a manner independent of the receptor role of Ihog in transducing the Hh signal. The Ihog-Ihog *trans*-homophilic binding site overlaps with the Ihog-Hh binding interface and requires the heparin-binding site, suggesting direct competition between the dual roles of Ihog proteins. By combining experimental data and mathematical modeling, we determined Hh binding to Ihog dominates and can disrupt pre-established Ihog-Ihog *trans*-homophilic interactions, resulting in Hh-Ihog complexes. Our results indicated that the weaker Ihog-Ihog *trans* interactions promote and stabilize membrane contacts along the cytonemes and the disruption of some of these interactions by the stronger Hh-Ihog interaction could contribute to cytoneme-mediated transport of Hh or internalization of the ligand-receptor complex. Thus, we proposed that the apparently

incompatible functions of Ihog in homophilic adhesion and ligand binding cooperate to assist Hh transport and reception along cytonemes.

## Results

### Ihog stabilizes cytonemes in a manner independent of Hh receptor function

The Hh receptor component Ihog localizes to cytonemes in the *Drosophila* wing imaginal discs and abdominal histoblasts. Increasing Ihog abundance makes cytonemes in the histoblasts less dynamic and enables easier microscopic detection of these structures in the wing disc (*Callejo et al., 2011*; *Bilioni et al., 2013*; *Bischoff et al., 2013*; *González-Méndez et al., 2017*). However, it is not clear how ectopic Ihog proteins influence the behavior and morphology of cytonemes. To explore the mechanism by which Ihog proteins affect cytonemes, we transiently expressed Ihog or the actin-binding domain of moesin fused to green fluorescent protein (GFP) (GMA-GFP) in the Hh-receiving cells in the A compartment using a *ptc-GAL4* driver in combination with *tub-GAL80*^ts. Cytonemes projecting from the *ptc-GAL4* expressing cells were examined in the 3rd instar larvae wing discs 24 hr after shifting to 29°C by staining with antibodies recognizing GFP or Ihog (*Figure 1A*). Unlike the short, mostly uniform cytonemes visualized by staining for GMA-GFP, the cytonemes with ectopic expression of Ihog were longer with periodic annular structures (*Figure 1B,C*). These annular structures were proposed to represent stable links between Hh-sending and Hh-receiving cytonemes (*González-Méndez et al., 2017*).

Several other Hh pathway components, including the Hh ligands, Ptc, and the *Drosophila* glypicans Division abnormally delayed (Dally) and Dally-like (Dlp), localize to cytonemes (*Ayers et al., 2010*; *Chen et al., 2017*; *González-Méndez et al., 2017*). To determine if these effects on cytoneme structures were unique to Ihog, we ectopically expressed each of these components

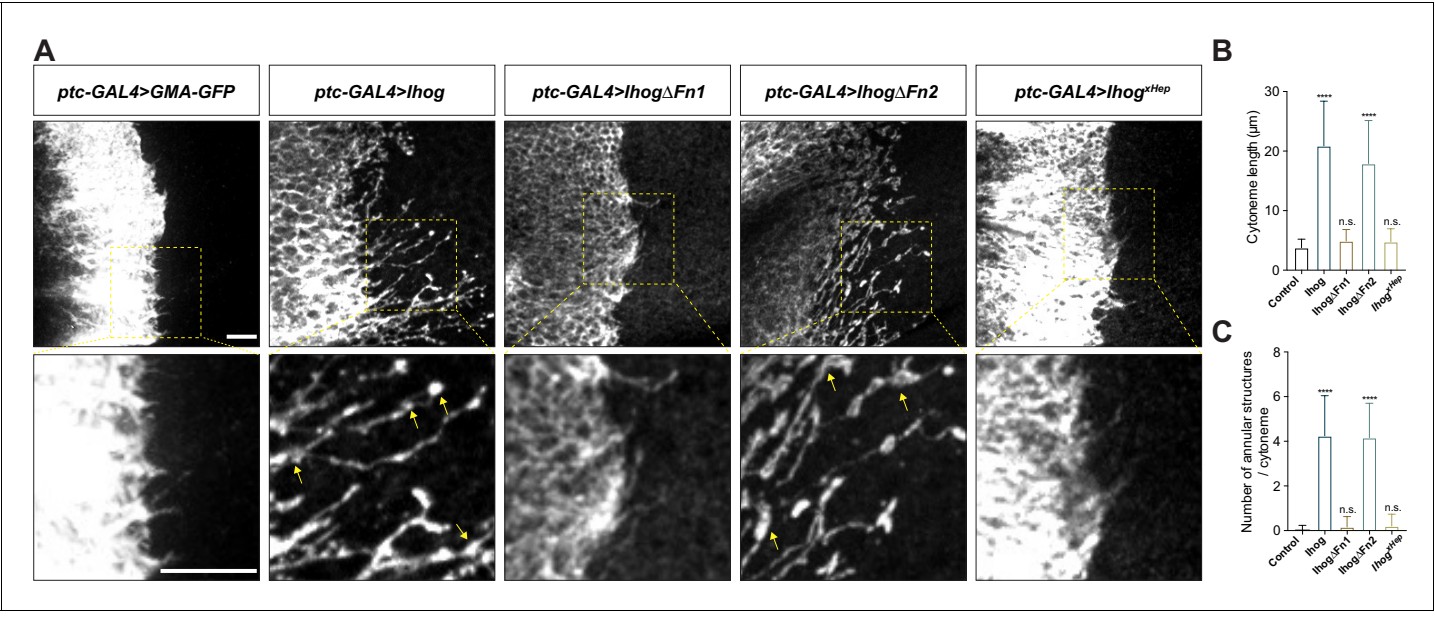

**Figure 1.** Ihog mediates cytoneme stabilization via the Fn1 domain. (**A**) Wing discs from 3rd instar larvae carrying *ptc-GAL4, tub-GAL80*^ts and the indicated *UAS*-transgenes were immunostained for GFP or Ihog to visualize cytonemes projecting from Hh-receiving cells. Yellow arrows indicate the annular structures along the cytonemes. Scale bar, 10 µm. (**B, C**) Quantification of the average cytoneme length (**B**) and the average annular structures number per cytoneme (**C**) in the wing disc. Each bar shows the mean ± SD (n > 30). One-way ANOVA followed by Dunnett's multiple comparison test was used for statistical analysis. ns, not significant. ****p<0.0001.

The online version of this article includes the following source data and figure supplement(s) for figure 1:

**Source data 1.** Contains numerical data plotted in *Figure 1B and C*.

**Figure supplement 1.** Ectopic expression of Ihog, but not other Hh pathway components, stabilizes cytonemes.

individually using the *ptc-GAL4* driver in the wing imaginal disc cells. For these experiments, we included *UAS-Myr-RFP*, which encodes a myristoylated form of red fluorescent protein, to mark the cell membrane and enable visualization of the cytonemes. Of the tested Hh components, only expression of Ihog lead to formation of the annular structures or increased cytoneme length (*Figure 1A-C*, *Figure 1—figure supplement 1*). We defined the increased cytoneme length and presence of annular structures as 'cytoneme stabilization'.

Previous biochemical and structural studies showed that the Ihog Fn1 domain is involved in binding to the Hh ligand via a heparin-binding surface (*McLellan et al., 2006*; *Yao et al., 2006*), whereas the Fn2 of Ihog interacts with Ptc. Both Fn1 and Fn2 domains are required for formation of a high-affinity multimolecular complex of Ihog with Ptc and Hh during Hh signal reception (*Zheng et al., 2010*). We performed a structure-function analysis by expressing Ihog variants lacking either the first FNIII domain (Fn1) (IhogΔFn1) or the second Fn2 (IhogΔFn2) or with mutations in the heparin-binding surface (Ihog$^{xHep}$) and quantified the cytoneme-stabilizing effects. These studies revealed that both the increased frequency of annular structures and length of the cytonemes required Fn1 and an intact heparin-binding surface (*Figure 1A-C*).

*ptc* not only encodes a component of the Hh receptor but is also a transcriptional target of Hh signaling (*Hooper and Scott, 1989*; *Nakano et al., 1989*; *Phillips et al., 1990*; *Capdevila et al., 1994*; *Tabata and Kornberg, 1994*). The highest expression of *ptc* is present in a narrow stripe of A cells adjacent to the A/P compartment boundary (Ptc$^{high}$). Anterior to the stripe, expression decreases over several cell diameters to a lower level which is maintained throughout the A compartment (Ptc$^{low}$). In contrast, *ptc* is not expressed in Hh-secreting P cells, which are insensitive to Hh stimulation (Ptc$^{neg}$). We generated randomly distributed Ihog-overexpressing cells throughout the A and P compartments in the wing imaginal discs. We observed stabilized cytonemes emanating from with Ihog-expressing clones located not only at the Ptc$^{high}$ A/P compartment boundary but also within the Ptc$^{low}$ A compartment and Ptc$^{neg}$ P compartment (*Figure 2A*, upper row). Thus, the cytoneme-stabilizing effect of Ihog was independent of Ptc, consistent with the ability of Ihog lacking the Fn2 domain to perform this function (*Figure 1*). Moreover, the expression of Ptc-binding-deficient IhogΔFn2 in the A/P boundary cells resulted in stable cytonemes projecting both posteriorly toward the Hh-secreting P cells and anteriorly away from the Hh source (*Figure 2B*). These observations indicated that Ihog stabilizes cytonemes through a mechanism different from that used for the formation of Ihog-Ptc receptor complex, which exhibits high-affinity binding to Hh ligands.

Ihog interacts with Hh both in the context of the Ihog-Ptc complex and independently of Ptc (*McLellan et al., 2006*; *Yan et al., 2010*; *Zheng et al., 2010*). We examined whether the Ptc-independent Ihog Fn1-Hh interaction contributes to Ihog-mediated cytoneme stabilization. In the wing imaginal discs, Hh is produced and secreted exclusively from the P compartment (Hh$^{high}$), from where it spreads only a few cell diameters into the A compartment (*Basler and Struhl, 1994*; *Capdevila et al., 1994*; *Tabata and Kornberg, 1994*). Cells of the A compartment do not themselves produce Hh and are thus referred to as Hh$^{neg}$. We reasoned that, if Ihog Fn1-mediated binding to Hh contributes to the cytoneme-stabilizing effect, cytonemes projecting from the Ihog-expressing wing disc cells in the Hh$^{neg}$ A or Hh$^{high}$ P compartment should display different properties. Consistent with Ptc-independent interactions occurring between Ihog and Hh, endogenous Hh accumulated along Ihog-overexpressing cytonemes either projecting from clones located within the P compartment or from boundary clones projecting posteriorly toward the Hh source (*Figure 2A*, lower row). No or very little Hh staining was detected with Ihog-expressing cytonemes from clones within the A compartment or with Ihog-expressing cytonemes from clones at the A/P boundary and extending into the A compartment. Despite the absence or limited amount of Hh ligands, Ihog expression stabilized all the cytonemes projecting within or toward the A compartment. These results indicated that Ihog Fn1-mediated binding to Hh ligands did not account for the stable cytonemes visualized by ectopic Ihog expression in the wing imaginal disc cells.

Quantification of cytoneme length and the number of annular structures per cytoneme for Ihog-expressing clones in the A compartment, P compartment, and at the boundary showed that cytoneme stabilization by Ihog was independent of position within the wing disc and thus the abundance of Ptc or Hh ligands (*Figure 2C,D*). We also quantified cytoneme-stabilizing properties for IhogΔFn2 in flip-out clones located close to the A/P compartment boundary, which also showed no difference between cytonemes projecting posteriorly towards the Hh-secreting P cells and those projecting anteriorly away from the Hh source (*Figure 2E,F*). Collectively, our results indicated that neither the



**Figure 2.** Ihog mediates cytoneme stabilization in a manner independent of the Hh receptor function. (**A**) Wing discs from 3rd instar larvae carrying flip-out clones expressing *UAS-Ihog* were immunostained for Ihog (green) and Ptc or Hh (red) as indicated. Dashed yellow line indicate the A/P compartment boundary; white arrows indicate clones located within the P compartment; blue arrows indicate cytonemes of clones located next to the A/P boundary that project toward the Hh producing cells; blue arrowheads indicate cytonemes of clones located next to the A/P boundary that project

*Figure 2 continued on next page*

*Figure 2 continued*
away from the Hh producing cells; yellow arrowheads indicate cytonemes from clones located within the A compartment. Scale bar, 20 μm. (**B**) Flip-out clones expressing *UAS-IhogΔFn2* viewed from the basal side at low magnification and in transverse, lateral, and basal sections of the same clone, showing localization of IhogΔFn2 proteins (immunostained for Ihog) at the lateral cell-cell contacts and basal cytonemes. Scale bar, 20 μm. (**C, D**) Quantification of the average cytoneme length and the average annular structure number per cytoneme for A (clones located in the Ptc^low A compartment), B (Ptc^high A/P compartment boundary), and P (Ptc^neg P compartment). (**E, F**) Quantification of the average cytoneme length and the average annular structure number per cytoneme for A (cytonemes projecting anteriorly away from the Hh source) and P (cytonemes projecting posteriorly toward the source of Hh). Each bar shows the mean ± SD (n > 30 clones). One-way ANOVA followed by Tukey's multiple comparison test (**C, D**) or the two-tailed unpaired t-test (**E, F**) was used for statistical analysis. ns, not significant.
The online version of this article includes the following source data for figure 2:

**Source data 1.** Contains numerical data plotted in *Figure 2C,D,E and F*.

presence of Ptc nor Hh is necessary for Ihog-mediated cytoneme stabilization. This function of Ihog was separate from its function in the Hh receptor complex.

## Ihog facilitates cytoneme stabilization through *trans*-homophilic binding supported by glypicans

Previously, we showed that the Ihog Fn1 domain not only plays an essential role in Hh signal transduction but also mediates cell-cell interactions in a homophilic manner (*Hsia et al., 2017*; *Wu et al., 2019*). Because our data indicated that Ihog stabilized cytonemes through the Fn1 domain in a manner independent of Hh receptor function (*Figures 1* and *2*), we hypothesized that Ihog Fn1-mediated homophilic *trans* interactions were responsible for cytoneme stabilization. The region that we identified as mediating the *trans* Ihog-Ihog interaction overlaps with the region that mediates the interaction with Hh on the Ihog Fn1 domain and includes a region where the negatively charged glycan heparin binds (*McLellan et al., 2006*; *Wu et al., 2019*). In vitro, heparin is required for not only Ihog-Hh binding but also Ihog-Ihog homophilic *trans* interactions (*McLellan et al., 2006*; *Wu et al., 2019*). Thus, a model for Ihog-Ihog homophilic *trans* interactions involves heparin-dependent bridging of positively charged surfaces on the two opposing Fn1 domains, in a manner similar to heparin-bridged Ihog-Hh interactions (*McLellan et al., 2006*; *Wu et al., 2019*).

Heparin used in previous in vitro assays is an intracellular glycosaminoglycan (GAG) that is not present on the cell surface or along the cytonemes. Thus, heparin is unlikely to mediate Ihog-Ihog *trans* interactions in vivo. Heparan sulfate, which is an extracellular GAG structurally related to heparin and ubiquitously located on the cell surface or in the surrounding extracellular matrix, was subsequently found to supply the function of heparin and mediate Ihog-Hh interaction in vitro (*Zhang et al., 2007*). Heparan sulfate is also covalently attached to proteins forming heparan sulfate proteoglycans (HSPGs), thus, heparan sulfate or HSPGs may serve as the physiological correlate of heparin to mediate Ihog-Ihog homophilic *trans* binding and Ihog-Hh binding. Dally and Dlp are two *Drosophila* glycosylphosphatidylinositol (GPI)-anchored HSPGs, which can be membrane-tethered or released from cells upon cleavage (*Bernfield et al., 1999*). Dally and Dlp are known to be involved in modulating the transport and reception of the Hh signal (*Lum et al., 2003*; *Lin, 2004*; *Tabata and Takei, 2004*; *Eugster et al., 2007*; *Ayers et al., 2010*; *Yan et al., 2010*; *Bilioni et al., 2013*). Ihog-expressing cytonemes rarely extend across large clonal populations of *dally* and *dlp* double mutant cells (*González-Méndez et al., 2017*), indicating that Dally and Dlp could be the major source of heparan sulfate that enables Ihog-Ihog homophilic *trans* interactions in vivo. Consistent with this hypothesis, we detected a striking accumulation of endogenous Dlp along Ihog-expressing cytonemes not only in the P compartment or along the A/P boundary where both Ihog-Hh and Ihog-Ihog interactions exist (*Figure 3A*) but also within the A compartment that lacks Hh and where only Ihog-Ihog homophilic interactions could occur (*Figure 3B*). Additionally, Dlp accumulated along the apical and lateral cell-cell contacts (*Figure 3B*; yellow outlined regions), where homophilic Ihog *trans* binding contributes to cell segregation in the wing imaginal disc epithelium (*Hsia et al., 2017*; *Wu et al., 2019*). We also observed that ectopic expression of Ihog caused the accumulation of endogenous Dally along the apical and lateral cell-cell contacts and basal cytonemes (*Figure 3—figure supplement 1*).

The Ihog-induced accumulation of Dally and Dlp along the basal cytonemes is consistent with a crucial contribution of either Dally or Dlp in Ihog-mediated cytoneme stabilization (*González-*



**Figure 3.** Ectopic Ihog induces accumulation of glypicans at lateral cell-cell contacts and along basal cytonemes. Wing discs from 3rd instar larvae carrying flip-out clones expressing the indicated *UAS*-transgene were immunostained for Ihog (green), Dlp (red), and Hh (blue). Dashed yellow lines indicate the A/P compartment boundary, which is determined by the expression of endogenous Hh; dashed blue lines indicates the dorsal/
*Figure 3 continued on next page*

*Figure 3 continued*

ventral (D/V) compartment boundary. (**A**) Ihog or Ihog mutants were expressed in wing discs. White arrows indicate clones located within the P compartment; blue arrows indicate cytonemes of clones located next to the A/P boundary that project toward the Hh producing cells; blue arrowheads indicate cytonemes of clones located next to the A/P boundary that project away from the Hh producing cells; yellow arrowheads indicate cytonemes from clones located within the A compartment. (**B**) Flip-out clones expressing *UAS-Ihog* viewed from the basal side at low magnification, showing their position relative to the A/P and D/V boundaries, and in lateral, transverse, and basal sections of the zoomed area. Blue outline indicates the flip-out clone flanking the D/V boundary; the yellow outline indicates the clone several cell diameters away from the D/V boundary. Scale bar, 20 μm.

The online version of this article includes the following figure supplement(s) for figure 3:

**Figure supplement 1.** Ectopic Ihog induces accumulation of Dally and Dlp that is associated with the different distributions of the two glypicans.

---

*Méndez et al., 2017*; *Simon et al., 2020*). However, ectopic expression of neither Dally nor Dlp resulted in cytoneme stabilization in the wing discs (*Figure 1—figure supplement 1*). Thus, the roles of Ihog and the glypicans in stabilizing the cytonemes are different. Given the known function of heparin as a bridging molecule in Ihog-Ihog or Ihog-Hh interactions, our results suggested that the heparan sulfate chains of Dally or Dlp provide this function in Ihog-Ihog *trans* interactions. Consistent with this model, the Ihog-induced Dally or Dlp accumulation reflected the different distributions of the two glypicans. Dlp is distributed in most cells, except in a zone ~7–10 cell diameters in width and centered at the dorsal ventral (D/V) boundary. Dally is also broadly distributed; however, Dally abundance is highest along the D/V boundary (*Fujise et al., 2001*; *Fujise et al., 2003*; *Han et al., 2005*). In agreement with these different distributions, both Dally and Dlp were enriched along the cytonemes and at the apical and lateral cell contacts of Ihog-expressing cells away from the D/V boundary, whereas the Ihog-expressing cells flanking the D/V boundary were positive for Dally with little or no detectable Dlp (*Figure 3B*, blue outline; *Figure 3—figure supplement 1*). Similar to the Ihog-mediated homophilic binding, Ihog-induced Dlp accumulation occurred in the absence of the Ihog Fn2 domain (*Figure 3A*). In contrast, neither ectopic expression of IhogΔFn1 nor Ihog$^{xHep}$, both of which lack homophilic binding capability, resulted in the accumulation of Dlp (*Figure 3A*). Taken together, these results indicated that Ihog Fn1-mediated homophilic *trans* interactions, assisted by the heparan sulfate chains of Dally or Dlp in the wing imaginal discs, contribute to cytoneme stabilization.

## Homophilic Ihog *trans* interactions promote direct cytoneme-cytoneme contact formation

We previously found that ectopic expression of Ihog in the non-adherent *Drosophila S2 cells induces cell* aggregation via homophilic *trans* interactions (*Wu et al., 2019*). Pairs of Ihog-YFP-positive S2 cells in close contact (aggregated) showed a peak of Ihog-YFP signal along the site of cell-cell contact (*Figure 4—figure supplement 1A–C*, *Figure 4—videos 1* and *2*). In contrast, Ihog-YFP was enriched in filopodia-like structures on dispersed S2 cells when evaluated for live cells or fixed cells (*Figure 4—figure supplement 1D–G*, *Figure 4—videos 3* and *4*). Because *Drosophila* S2 cell filopodia recapitulate structural and functional characteristics of cytonemes in the imaginal disc (*Bodeen et al., 2017*), here, we take advantage of these dispersed Ihog-YFP-expressing S2 cells to evaluate the possibility of filopodia-localized Ihog proteins in participating homophilic *trans* interactions. We examined the behavior of these Ihog-positive filopodia between an Ihog-positive cell and an Ihog-negative cell and between pairs of Ihog-positive cells. These structures were found at regions where two Ihog-positive cells were in close apposition with the filopodia projecting toward the adjacent Ihog-positive cell (*Figure 4—figure supplement 2A*). In contrast, an Ihog-positive cell extend fewer filopodia toward an Ihog-negative cell (*Figure 4—figure supplement 2A,B*). Two closely positioned Ihog-positive cells exhibited an increase in the number of filopodia oriented toward the nearby Ihog-positive cell (*Figure 4—figure supplement 2C*). Moreover, using live-cell time-lapse imaging, we observed that filopodia extending from non-adjacent Ihog-YFP-expressing cells interdigitated, then shortened to bring the two cells closer, and finally established a stable cell-cell contact (*Figure 4—figure supplement 3*). Collectively, the observations in S2 cells suggested

that filopodia-localized Ihog proteins engaged in homophilic *trans* binding evidenced by the contact initiation along the filopodia of non-adjacent Ihog-expressing S2 cells.

To explore if such events occurred in vivo, we generated Ihog-expressing clones in the wing imaginal discs and found that cytonemes projecting from closely positioned clones appeared to come into contact (*Figure 4A*; arrows). Unlike cultured S2 cells, the wing imaginal disc epithelial cells tightly adhere to their immediate neighbors and maintained stable cell-neighbor relationships (*Garcia-Bellido et al., 1973*; *Gibson et al., 2006*). Cytoneme-cytoneme interactions are unlikely to lead to epithelial cell rearrangements, thus a reduction in the distance between non-adjacent Ihog-expressing clones could not be used as the functional readout of Ihog-Ihog *trans*-interaction along the cytonemes. Therefore, we examined whether direct membrane contacts were established along Ihog-localized cytonemes.

Membrane contacts are typically separated by less than 100 nm of extracellular space, which is below the resolution of conventional light microscopy. To examine whether membrane contacts were established along Ihog-stabilized cytonemes, we combined the CoinFLP-LexGAD/GAL4 system and the GFP Reconstitution Across Synaptic Partners (GRASP) system (*Feinberg et al., 2008*; *Gordon and Scott, 2009*; *Bosch et al., 2015*). CoinFLP-LexGAD/GAL4 allows generation of tissues composed of clones that express either GAL4 or LexGAD, thus enabling the study of interactions between different groups of genetically manipulated cells (*Bosch et al., 2015*). In the GRASP system, two complementary parts of a 'split GFP' (spGFP$_{1-10}$ and spGFP$_{11}$) are fused to the extracellular domains of mouse CD4, one under UAS control and the other under LexAop control. Although individually the membrane-tethered spGFP fragments are not fluorescent, reconstitution of GFP generates fluorescence at the boundary of immediately adjacent clones that express the complementary spGFP fragments (*Bosch et al., 2015*). We expressed RFP-tagged Ihog together with CD4-spGFP$_{11}$ and HA-tagged IhogΔFn2 with CD4-spGFP$_{1-10}$ (*Figure 4B*, *Figure 4—video 5*). With this system, we monitored cells for the presence of either Ihog or IhogΔFn2 using an antibody recognizing Ihog and cells positive for only IhogΔFn2 using an antibody against the HA tag.

As expected from the CoinFLP system, clones expressing Ihog-RFP plus CD4-spGFP$_{11}$ and those expressing IhogΔFn2-HA plus CD4-spGFP$_{1-10}$ were randomly distributed in the wing imaginal discs. When these two types of clones were located immediately adjacent to each other, GRASP fluorescence was detected both at the lateral cell-cell contacts and along the basal cytonemes of the adjacent clones expressing Ihog-RFP plus CD4-spGFP$_{11}$ or IhogΔFn2-HA plus CD4-spGFP$_{1-10}$ (*Figure 4C,D*; blue outline and blue arrow). The basal cytonemes emanating from wing imaginal disc cells can reach as far as several cell diameters, thus, if cytoneme-localized Ihog proteins participate in *trans*-homophilic binding, direct cytoneme-cytoneme contacts from non-adjacent cells could be preferentially established among these cytonemes expressing ectopic Ihog proteins (*Figure 4B*). Indeed, GRASP fluorescence also appeared along the length of the cytonemes projecting from the non-adjacent Ihog-RFP plus CD4-spGFP$_{11}$ and IhogΔFn2-HA plus CD4-spGFP$_{1-10}$ expressing cells that do not share common boundaries as indicated by the lack of GRASP fluorescence throughout the lateral clonal borders (*Figure 4C,D*; yellow outline and yellow arrow). Therefore, the GRASP-marked cytoneme-cytoneme contacts from non-adjacent clones suggested that membrane contacts were initiated and established along Ihog-expressing cytonemes, supporting the idea that Ihog-Ihog *trans*-binding can occur along opposing cytoneme membranes in vivo.

## Computational modeling predicts that homophilic Ihog *trans* interactions increase cytoneme length and bundling

We developed a stochastic model to investigate the influence of the homophilic *trans* interaction strength on the dynamics of cytonemes. In the model, cytonemes were represented as filamentous structures with variable numbers of discrete segments. We considered the elongation, shrinkage, translocation, and interaction events of cytonemes on the cell surface: An elongation or shrinkage event was represented by the addition or removal of one segment to or from an existing cytoneme; a translocation event was represented as the movement of a cytoneme along the cell surface; and interaction events of two cytonemes in contact were represented by pairwise interactions between segments.

We set the elongation probability of the $i^{th}$ cytoneme to exponentially decay with its length:

$$p_{elongation,i} = p_{elongration}^{0} \times e^{-\alpha \cdot x_i} \tag{1}$$



**Figure 4.** Homophilic Ihog-Ihog *trans* binding enables direct cytoneme-cytoneme contact formation. (**A**) Wing imaginal discs from 3rd instar larvae carrying flip-out clones expressing *UAS-Ihog* were immunostained with antibodies against Ihog (green), Ptc (blue), and GFP (red) as indicated. Yellow arrows indicate Ihog-enriched cytonemes projecting from closely positioned clones. Scale bar, 20 μm. (**B**) Diagram illustrating cytoneme-cytoneme contact between non-adjacent clones expressing ectopic Ihog or IhogΔFn2 that are capable of homophilic *trans* binding. The green color corresponds to the GRASP signal, which is not only detected at the lateral contacts and along the basal cytonemes of the adjacent clones, but also along cytonemes projecting from non-adjacent clones that express CD4-spGFP$_{1-10}$/IhogΔFn2 and CD4-spGFP$_{11}$/Ihog. (**C, D**) Lateral and basal sections of a wing imaginal disc from 3rd instar larvae carrying clones marked by the CoinFLP-LexGAD/GAL4 system and the GRASP system as indicated. The wing discs were immunostained with antibodies against Ihog (red, both Ihog-RFP and IhogΔFn2-HA expressing cells) and HA (blue, IhogΔFn2-HA expressing cells) as indicated. GRASP signal is green. Blue outlines indicate clones expressing CD4-spGFP$_{1-10}$ and IhogΔFn2-HA that are immediately adjacent to clones expressing CD4-spGFP$_{11}$ and Ihog-RFP; yellow outlines indicate CD4-spGFP$_{1-10}$ and UAS- IhogΔFn2-HA clones that are distant from those expressing CD4-spGFP$_{11}$ and Ihog-RFP. Blue arrowheads indicate GFP fluorescence along the lateral side of the outlined clones. Yellow arrowheads indicate absence of GFP fluorescence along the lateral sides of the outlined clones. (**D**) Blue and yellow arrows indicate GFP fluorescence along the length of the cytonemes projecting from the clones indicated by blue and yellow outlines (**C**), respectively. Scale bar, 20 μm.

The online version of this article includes the following video and figure supplement(s) for figure 4:

**Figure supplement 1.** Subcellular localization of Ihog proteins in live S2 cells.

**Figure supplement 2.** Ihog is enriched in filopodia-like structures of closely positioned Ihog-expressing cells.

**Figure supplement 3.** Cell-cell contact initiation along the filopodia of non-adjacent Ihog-expressing S2 cells.

**Figure 4—video 1.** Subcellular localization of Ihog proteins in aggregated live S2 cells.

https://elifesciences.org/articles/65770#fig4video1

**Figure 4—video 2.** Subcellular localization of Ihog proteins in aggregated live S2 cells.

https://elifesciences.org/articles/65770#fig4video2

**Figure 4—video 3.** Subcellular localization of Ihog proteins in singular live S2 cells.

*Figure 4 continued on next page*

*Figure 4 continued*

https://elifesciences.org/articles/65770#fig4video3

**Figure 4—video 4.** Subcellular localization of Ihog proteins in singular live S2 cells.

https://elifesciences.org/articles/65770#fig4video4

**Figure 4—video 5.** Homophilic Ihog-Ihog *trans* binding enables direct cytoneme-cytoneme contact formation.

https://elifesciences.org/articles/65770#fig4video5

where $p^0_{elongation}$ is the base elongation probability at the cell surface, $\alpha$ is the decay coefficient, and $x_i$ is the number of segments in the $i^{th}$ cytoneme (that is the length that the cytoneme extended from the cell surface). This decay relationship represents the increasing difficulty in transporting materials to the tip of the cytoneme as elongation occurs and increasing difficulty in the occurrence of elongation as the membrane tension increases, thereby resisting elongation.

The shrinkage rate at the tip of the $i^{th}$ cytoneme is modeled as:

$$p_{shrink,i} = p^0_{shrink} \times e^{-T_i \cdot E_{ii}} \tag{2}$$

where $p^0_{shrink}$ is the intrinsic shrinkage rate without any homophilic *trans* interaction, $E_{ii}>0$ is the strength of the homophilic *trans* interaction between a pair of segments, and $T_i$ is the number of neighboring cytonemes with which this $i^{th}$ tip segment interacts. Thus, the homophilic *trans* interactions at the tip segment represent additional energy barriers to a shrinkage event.

To enhance simulation efficiency, we employed the quasi-equilibrium approximation (*Goutsias, 2005*) to simulate the pairwise interactions between segments on neighboring cytonemes. First, we computed the probability of establishing a homophilic *trans* interaction between a pair of neighboring cytoneme segments:

$$p_{interaction} = \frac{\exp(E_{ii})}{1 + \exp(E_{ii})} \tag{3}$$

Based on $p_{interaction}$, we randomly assigned a binary state variable, $s^{i,j}_k$ (0 for not interacting, 1 for interacting), to the $k^{th}$ pair of neighboring segments in the $i^{th}$ and $j^{th}$ cytonemes for each simulation step. Thus, the total homophilic interactions of the $i^{th}$ cytoneme is calculated as:

$$E^i_{interaction} = E_{ii} \times \sum_{j \in neighbor \ of \ i} \sum_k s^{ij}_k \tag{4}$$

The cytonemes can also translocate along the cell periphery. A translocation event involves breaking the existing homophilic *trans* interactions and establishing new homophilic *trans* interactions. We computed the differences in the *trans* interactions before and after a possible translocation event for the $i^{th}$ cytoneme as $\Delta E^i_{translocation}$. Using the computed probabilities and energy differences, we performed stochastic simulations and collected the cytoneme configurations after the simulations reached steady state (see Materials and methods for details).

We simulated the system with no ($E_{ii} = 0$) homophilic *trans* interactions (*Figure 5A*, left) and with homophilic *trans* interactions of moderate strength ($E_{ii} = 15$) (*Figure 5A*, right). Without any homophilic *trans* interactions, the simulations resulted in much fewer numbers of segments (shorter cytoneme length). For $E_{ii} = 15$, the simulations predicted more variability in the length of cytonemes than was predicted at $E_{ii} = 0$. By capturing 1001 snapshots from the random simulations for $E_{ii} = 0$ and 15, we found that the simulations produced cytoneme lengths that were significantly longer at $E_{ii} = 15$ (*Figure 5B*). Additionally, the number of established pairwise interactions between cytonemes greatly increased at $E_{ii} = 15$ (*Figure 5C*). Thus, the simulations indicated that cytoneme length correlated with the number of cytoneme-cytoneme interaction events (*Figure 5D*, Pearson r = 0.7939). By varying the strength of homophilic *trans* interactions, we also found that average cytoneme length increased with stronger cytoneme-cytoneme interactions (*Figure 5E*).

In the snapshots of the simulations, we observed extensive pairwise interactions among adjacent cytonemes only when we set $E_{ii}>0$ (*Figure 5A*, arrows). We defined this phenomenon as 'cytoneme bundling' and quantified this phenomenon with a cytoneme bundling index (see Materials and methods). The cytoneme bundling index increased as the strength of the homophilic



**Figure 5.** Computational modeling predicts that *trans* homophilic Ihog interactions stabilize cytonemes. (**A**) Snapshots of simulated cytoneme configurations with no (left, $E_{ii} = 0$) homophilic *trans* interactions and with moderately strong (right, $E_{ii} = 15$) homophilic *trans* interactions. The solid black horizontal lines at the bottom represent the cell surface. The blue vertical filaments are cytonemes, within which the elliptical elements are the individual segments. The red dots are the established pairwise interactions between neighboring segments. The orange arrows indicate the bundled neighboring cytonemes with extensive pairwise contacts. (**B, C**) The average cytoneme length and the number of cytoneme-cytoneme interactions at $E_{ii} = 0$ and 15. Each dot is obtained from 1 randomly picked snapshot from the simulation. Each bar shows the mean ± SD, n = 1001. (**D**) Correlation

*Figure 5 continued on next page*

*Figure 5 continued*

between the cytoneme length and the number of pairwise interactions at $E_{ii} = 15$. Each dot represents the length and the number of interactions for individual cytonemes. The line shows the best fit linear regression (Pearson r = 0.7939). (E) Effect of homophilic *trans* interaction strength on the average cytoneme length. The average length of the simulated cytonemes is plotted against $E_{ii}$ ranging from 0 to 50. Each bar shows the mean ± SD, n = 1001. (F) Effect of homophilic *trans* interaction strength on the formation of cytoneme bundles. The cytoneme bundling index for all cytoneme bundles identified from n=1001 random snapshots, each containing 30 cytonemes, were plotted against $E_{ii}$ ranging from 0 to 50. Each bar shows the mean ± SD, n = 1001. The two-tailed unpaired t-test (B, C) or one-way ANOVA followed by Sidak's multiple comparison test (E, F) was used for statistical analysis. ***p < 0.001, ****p < 0.0001. (G) Wing discs from 3rd instar larvae carrying *ptc-GAL4, tub-GAL80^ts*, and *UAS-GPI-YFP; UAS-Ihog* were immunostained for YFP (GPI, green) and Ihog (Ihog, red), followed by imaging with Airyscan. Yellow arrows indicate the annular structures observed by regular confocal microscopy; blue arrows indicate likely single cytonemes; red arrows indicate bundles containing multiple cytonemes. Scale bar, 5 µm.

The online version of this article includes the following source data and figure supplement(s) for figure 5:

**Source data 1.** Contains numerical data plotted in *Figure 5B,C,D,E and F*.
**Figure supplement 1.** Effect of homophilic *trans* interaction strength on the frequency of cytoneme bundle formation.
**Figure supplement 2.** The Ihog Fn1 domain is essential for cytoneme bundling.
**Figure supplement 3.** Loss of *Ihog* and its close paralogue *boi* from the wing imaginal discs cells resulted in cytonemes with reduced length.

*trans* interactions among cytonemes increased (*Figure 5F*). Furthermore, with increasing cytoneme-cytoneme homophilic *trans* interaction strength $E_{ii}$, we observed a decreased proportion of singular cytonemes and an increased proportion of cytonemes within the cytoneme bundles (*Figure 5—figure supplement 1*). These results predicted that cytonemes of Ihog-overexpressing cells would form extensive contacts and appear as bundles. By regular confocal microscopy, we observed an increase in annular contact sites, but we did not detect clear evidence of cytoneme bundles in the Ihog-over-expressing wing disc. This is likely because the diameter of cytonemes [100–200 nm (*Kornberg, 2014*; *Mattila and Lappalainen, 2008*)] is much less than the resolution limit (~250 nm laterally) of confocal microscopy. We then used Airyscan technology, which has a lateral resolution of 120 nm (*Huff, 2015*), to test the prediction of Ihog-induced bundling of cytonemes in the wing disc. We imaged Ihog-expressing cytonemes in wing discs cells co-labeled with membrane marker glycosyl-phosphatidyl-inositol–YFP (*Greco et al., 2001*) (GPI–YFP, *Figure 5G*). Consistent with the computational modeling prediction, we detected thin cytonemes (*Figure 5G*, blue arrows) that appeared to form thick bundles (*Figure 5G*, red arrows). In contrast, we rarely observed cytoneme bundling upon expression of only the membrane marker GPI-YFP or expression of the homophilic binding-deficient Ihog variants IhogΔFn1 and Ihog^xHep (*Figure 5—figure supplement 2*). Therefore, the experimental observations are consistent with the computational modeling predictions that the augmented cytoneme-cytoneme interactions mediated by ectopic Ihog lead to bundled cytonemes.

Taken together, our in vivo and *in silico* studies showed that *trans* Ihog-Ihog homophilic interactions increase cytoneme length and bundling (*Figures 1–5*), which in turn explained why the cytonemes expressing ectopic Ihog proteins were much easier to be detected microscopically and Ihog was often used as a cytoneme marker to visualize these structures (*Callejo et al., 2011*; *Bilioni et al., 2013*; *Bischoff et al., 2013*; *González-Méndez et al., 2017*; *Portela et al., 2019*; *González-Méndez et al., 2020*). It is also worth noting, the large cytoneme bundles were only observed when Ihog was overexpressed, and physiological amounts of Ihog were not sufficient to promote extensive bundling of cytonemes detectable under regular or Airyscan confocal microscope (*Figure 1A*, *Figure 1—figure supplement 1*, *Figure 5—figure supplement 2*). Nevertheless, consistent with the finding that ectopic Ihog leads to elongated cytonemes, knockdown of *ihog* in the absence of its close paralog-encoding gene *boi* resulted in cytonemes with significantly reduced length compared with the length of cytonemes in *boi* mutant animals retaining normal amounts of Ihog (*Figure 5—figure supplement 3*). Therefore, endogenous Ihog proteins also played a crucial role in regulating the structure of cytonemes, which our data indicated involved Ihog-mediated cytoneme-cytoneme interactions.

## Heterophilic Ihog-Hh binding dominates over homophilic Ihog *trans* interaction

A single Ihog protein can participate in either an Ihog-Ihog *trans* interaction or an Ihog-Hh interaction; therefore, a single Ihog protein can mediate either its cytoneme-stabilizing function or its

ligand-binding function, but not both simultaneously. The dissociation constant for the nonlipid-modified recombinant HhN and the extracellular portion of Ihog containing the Fn1 and Fn2 domains (IhogFn1-2), measured in solution, is ~2 µM (*McLellan et al., 2006*; *Zhang et al., 2007*). Additionally, soluble HhN at 30 µM competes for Ihog homophilic interactions in a pull-down assay in which HhN and Ihog are mixed simultaneously (*Wu et al., 2019*). To explore the interaction hierarchy of Ihog-Ihog *trans* interactions and Ihog-Hh interactions, we used the S2 cell system. Homophilic *trans* interactions mediated by ectopically expressed Ihog result in aggregation of the normally non-adhesive S2 cells (*Hsia et al., 2017*; *Wu et al., 2019*). We performed Ihog-mediated cell aggregation assays with two populations of cells, one expressing Ihog and GFP and the other expressing Ihog and monomeric Cherry (mCherry), in the presence of exogenously applied recombinant HhN. Unexpectedly, even at 30 µM, a concentration 10 times higher than the reported IhogFn12-HhN dissociation constant, soluble HhN had little effect on Ihog-mediated cell aggregation (*Figure 6—figure supplement 1*). One explanation is that soluble HhN does not reach a sufficiently high concentration at the cell surface to compete for the extensive homophilic *trans* interactions that can be mediated by membrane-tethered Ihog proteins. Thus, we developed an assay to test the ability of plasma membrane-associated Hh to interfere with Ihog-Ihog *trans* interactions. Expression of cDNA encoding full-length Hh in S2 cells generates dually lipid-modified Hh ligands, whereas expressing cDNA encoding the amino-terminal signaling fragment results in HhN lacking a cholesterol modification (*Burke et al., 1999*; *Porter et al., 1995*). Although both forms of Hh ligands are competent to bind to the receptors and induce Hh signaling in ligand-receiving cells, HhN does not require Dispatched (Disp) for release from the producing cell. Thus, in S2 cells without also ectopically expressing Disp, only dually lipid-modified Hh ligands, but not HhN lacking a cholesterol modification, was associated with the membrane of the transfected cells (*Figure 6—figure supplement 2*). Using these cells, we assessed the relative strengths of Ihog-mediated ligand binding and homophilic *trans* interactions.

A heterogeneous aggregation of cells exhibits distinct morphological patterns when the relative strengths of the heterotypic and homotypic cell-cell adhesions differ. For example, a checkerboard-like pattern can occur when heterotypic cell-cell adhesions dominate (*Honda et al., 1986*). Therefore, we hypothesized that the morphological patterns produced by the aggregated Hh- and Ihog-expressing cells reflect the relative affinity of Ihog-Hh (heterotypic) and Ihog-Ihog (homotypic) interactions. To test this hypothesis, we prepared S2 cells expressing Hh or HhN and cells expressing Ihog along with either GFP or mCherry, mixed the dissociated cells, and assessed the pattern of the aggregated clusters. We found that Hh-expressing cells, which did not aggregate by themselves (*Figure 6A*, mCherry+Hh, GFP+Hh), aggregated when mixed with Ihog-expressing cells (*Figure 6A*, mCherry+Ihog, GFP+Hh). In contrast, HhN-expressing cells neither aggregated themselves (*Figure 6A*, mCherry+HhN, GFP+HhN) nor with Ihog-expressing cells (*Figure 6A*, mCherry+Ihog, GFP+HhN).

We compared the patterns of the cells in the clusters containing only Ihog-expressing cells and those containing both Ihog-expressing cells and Hh-expressing cells. We observed a checkerboard-like pattern with evenly distributed red and green cells in aggregates formed by cells expressing Hh with GFP and cells expressing Ihog with mCherry (*Figure 6B*). Most center cells within the cell aggregates had four or five neighbors (*Figure 6C*). Furthermore, among those neighbors, cells expressing the same transfected proteins and thus of the same color ('like' cell) were rare (*Figure 6B,D*). In contrast, aggregates of Ihog-expressing cells labeled with either GFP or mCherry exhibited a honeycomb pattern (*Figure 6B*): Most center cells had five or six neighbors (*Figure 6C*), and ~50% were 'like' cells (*Figure 6B,D*). For each aggregation assay, we confirmed by immunoblotting that transfected cells from the same experiment used for the aggregation assays expressed comparable amounts of Ihog and Hh proteins (*Figure 6—figure supplement 3*). Therefore, the different cellular patterns formed by cells expressing Hh and cells expressing Ihog versus those formed by differentially labeled Ihog-expressing cells suggested that the heterophilic interaction between Ihog and Hh is stronger than the homophilic *trans* interaction between Ihog proteins on an opposing cell surface.



**Figure 6.** Heterophilic binding of Ihog to Hh dominates over Ihog-mediated homophilic *trans* interactions. (**A**) S2 cells were transfected with plasmids expressing Ihog, Hh, or HhN along with expression plasmids for either GFP or mCherry as indicated. Cells were dissociated by trypsin treatment and then mixed for 12 hr to allow aggregation to occur. The top and middle rows show the mixing of cells expressing only the fluorescent proteins or together with Hh, HhN, or Ihog. Scale bar, 100 µm. The bottom row shows the indicated zoomed area from the middle row images. Scale bar, 50 µm.
*Figure 6 continued on next page*

*Figure 6 continued*

(B) Representative examples used for quantification of cell patterns in aggregates. The center cells in cell aggregates are indicated by purple asterisks. The neighboring cells are counted and labeled with yellow numbers. (C, D) The average numbers of total neighbor cells and 'like' (expressing the same proteins and thus the same color as the center cell) neighbor cells were quantified. Each bar shows the mean ± SD, n = 30 center cells (from n > 3 experiments). The unpaired two-tailed t-test was used for statistical analysis. ****p<0.0001.

The online version of this article includes the following source data and figure supplement(s) for figure 6:

**Source data 1.** Contains numerical data plotted in *Figure 6C and D*.
**Figure supplement 1.** Ihog-mediated homophilic *trans* interaction in S2 cells occurs in the presence of recombinant HhN.
**Figure supplement 2.** Subcellular localization of Hh and HhN in S2 cells.
**Figure supplement 3.** Comparison of the amount of Ihog and Hh in S2 cells analyzed in the cell-mixing assay.

## Computational modeling estimates the difference in strength between the heterophilic Ihog-Hh and homophilic Ihog *trans* interaction

Directly determining the affinities of the *trans* homophilic Ihog-Ihog and heterophilic Ihog-Hh interactions is difficult because Ihog-Ihog homophilic interactions could occur both in *trans* and in *cis* (i.e. interactions between Ihog protein on different vs. the same membrane) (*McLellan et al., 2006*; *Wu et al., 2019*). We thus took a computational approach to estimate the relative affinities of the heterophilic Ihog-Hh and homophilic Ihog *trans* interaction. Motivated by the observations that cells expressing Hh and Ihog produced a different pattern from the cells expressing Ihog, we estimated the difference in strength between the heterophilic Ihog-Hh and homophilic Ihog-Ihog *trans* interactions by modeling these interactions using a vertex-based *in silico* assay (*Bi et al., 2015*; *Park et al., 2016*). We explicitly included heterogeneous cell composition in our model in the following manner: The cells were approximated by polygons that can freely change their locations and shapes. Consequently, two interacting cells were represented by two polygons sharing a common edge. This interaction leads to an energy reduction, the magnitude of which depends on various properties including the strength of the cell-cell adhesive interactions. From the cell shapes and configurations of neighboring cells, mechanical energy ($e_i$) was calculated for each cell according to *Farhadifar et al., 2007* as:

$$e_i = \alpha \times (A_i - A_0)^2 + \beta \times P_i^2 - \frac{1}{2} \times \sum_{j \in neighboring\ cells} \gamma_{q_i q_j} \times l_{ij} \qquad (5)$$

The first term is the areal elasticity, which is represented by $\alpha$ (the elastic coefficient), $A_i$ (the area of the ith cell), and $A_0$ (the preferred cell area). The second term is the contractile energy, which is represented by $\beta$ (the contractile coefficient) and $P_i$ (the perimeter of the ith cell). The third term is the net adhesive energy between the ith cell and its neighbors, where $\gamma_{q_i q_j}$ is the line density of the adhesive energy between cell types $q_i$ and $q_j$, and $l_{ij}$ is the length of the common edge between the two cells. We have $\gamma_{II}$, $\gamma_{IH}$, and $\gamma_{HH}$, depending on the types of surface proteins expressed by the cells: both expressing Ihog, *II*, or one expressing Ihog and one expressing Hh, *IH*. Here, $\gamma_{HH} = 0$, because we did not observe cells both expressing Hh in contact with each other in the aggregation assays.

We used the Monte Carlo method (*Metropolis et al., 1953*) to simultaneously simulate 100 cells within a two-dimensional space. Gaps between cells were simulated as empty polygons that do not contribute to the mechanical energy of the system. With this system, the aggregation or segregation of cells is governed by $\sum e_i$. As a control, we simulated 100 Ihog-expressing cells with half labeled red and half labeled green, which produced a honeycomb morphological pattern (*Figure 7A*), within which a given center cell had 5.8 ± 0.6 neighbors, and 2.4 ± 0.9 of them had the same color label as the center cell (*Figure 7C*, I~I bars).

We simulated 50 Ihog-expressing cells and 50 Hh-expressing cells (*Figure 7B*). When we altered the ratio of the heterotypic and the homotypic interaction strength ($\gamma_{II}$, $\gamma_{IH}$ values), the morphology of the mixed system changed. From these values and the patterns, we obtained the average number and type of neighbor cells for any given center cell. We found that when $\gamma_{IH}$ is 30 times larger than $\gamma_{II}$, the mixed system exhibited the checkerboard-like morphological pattern (*Figure 6B*), within which each center cell had 4.1 ± 0.7 neighbors, and only 0.3 ± 0.5 of them were 'like' cells (*Figure 7C*, I~H bars).



**Figure 7.** *In silico* simulation estimates the difference in strength between the heterophilic Ihog-Hh and homophilic Ihog-Ihog *trans* interactions. (**A, B**) Representative steady-state patterns of the multicellular system from simulations with differentially labeled Ihog-expressing cells (**A**) and mixed Ihog- and Hh-expressing cells (**B**). (**C**) The average numbers of total neighbor cells and 'like' neighbor cells were quantified for scenarios (A, I~I bars) and (B, I~H bars). Data were obtained from 300 random snapshots. Each error bar shows the mean ± SD. (**D, E**) Blue lines are the quantified relationships

*Figure 7 continued on next page*

*Figure 7 continued*

between the average numbers of total neighbor cells (D) and 'like' neighbor cells (E) in a mixed system as a function of the difference in strength between the heterophilic Ihog-Hh ($\gamma_{IH}$) and homophilic Ihog-Ihog ($\gamma_{II}$) *trans* interactions. For comparison, the red dashed lines mark the values obtained from homogeneous system with differentially labeled Ihog-expressing cells. The shaded areas outline the standard deviations around the corresponding central average values. Each data point was calculated using 300 random snapshots from the simulation. (F) Simulation with $\gamma_{IH}:\gamma_{II} = 30$, starting with an aggregate formed from 50 Ihog-expressing cells, then 50 Hh-expressing cells were added into the simulation space (left). The energy-based evolution leads to the surface engagement of Hh-expressing cells onto the Ihog-expressing cell aggregate (middle) and eventually a checkerboard-like morphological pattern (right) appears as the simulation reaches steady state. (G) S2 cells were transfected with plasmids expressing Hh and GFP or Ihog and mCherry as indicated. Forty-eight hr after transfection, cells were resuspended by pipetting and then mixed as indicated. Cell mixture was incubated with gentle rotating for 12 hr. At the indicated time points, an aliquot of cells was removed and imaged with a confocal microscope. Representative images from three experiments are shown. Scale bar, 50 µm.

The online version of this article includes the following source data for figure 7:

**Source data 1.** Contains numerical data plotted in *Figure 7C*.

This simulation study of the effect of the parameter $\gamma_{II}: \gamma_{IH}$ predicted that the number and type of neighbor cells are sensitive to the $\gamma_{IH}:\gamma_{II}$ ratio. A honeycomb-to-checkerboard transition occurred when $\gamma_{IH}:\gamma_{II}$ is between 20 and 30 (*Figure 7D, E*). Therefore, the neighbor cell analysis from the experimental aggregation assay and the computational modeling suggested that the relative affinity of the Ihog-Hh heterotypic interaction is at least ~ 20–30 times higher than that of the Ihog-Ihog homotypic *trans* interaction. Previous sedimentation velocity and sedimentation equilibrium experiments predicted that the dimerization constants for the extracellular portion of Ihog comprising Fn1–2 range from 60 to 430 µM and that the dissociation constants for the interaction between HhN and this IhogFn1–2 region were between 0.4 and 8.0 µM (*McLellan et al., 2006*; *Zhang et al., 2007*). Although IhogFn1-2 dimerization in vitro could occur both in *cis* and in *trans* (*McLellan et al., 2006*; *Wu et al., 2019*), our modeling prediction of an affinity for the Ihog-Hh heterotypic interaction that is at least 20 times higher than that for the Ihog-Ihog homotypic *trans* interaction is consistent with the values obtained in the biochemical experiments (*McLellan et al., 2006*; *Zhang et al., 2007*).

## Heterophilic Ihog-Hh binding displaces pre-established homophilic Ihog *trans* interactions

Differential adhesion has been proposed to promote cell rearrangements in cell aggregates such that strongly adhesive cells sort together and weakly adhesive cells are excluded (*Foty and Steinberg, 2005*; *Steinberg, 2007*). Thus, from our computational analysis of interaction strength, we predicted that mixing Hh-expressing cells with pre-existing aggregates of Ihog-expressing cells would displace the relatively weaker Ihog-mediated homophilic interactions by establishing stronger Hh-Ihog interactions and result in a checkerboard pattern. To test this, we performed both computational model simulations and experiments with S2 cells. With $\gamma_{IH}:\gamma_{II}=30$ in the computational model, we observed the development of the checkerboard pattern as the simulation reached steady state (*Figure 7F*).

In the S2 cell experiment, we mixed differentially labeled Hh-expressing cells with preformed aggregates of Ihog-expressing cells (*Figure 7G*). Within 30 min after cell mixing, Hh-expressing cells were found on the surface of the pre-existing aggregates of Ihog-expressing cells, which we interpreted as Hh binding to Ihog proteins that were not engaged in homophilic adhesion. Over time, Hh-expressing cells were found inside the Ihog-expressing cell aggregates, such that, by 12 hr after mixing, all cell aggregates contained similar numbers of Hh-expressing cells and Ihog-expressing cells arranged in a checkerboard-like pattern (*Figure 7G*). This pattern is consistent with cell rearrangements caused by differential adhesion with the Hh-Ihog interaction exhibiting a higher affinity than the Ihog-Ihog homophilic *trans* interaction. Moreover, the observed cellular rearrangement indicated the Hh ligand disrupts *trans* Ihog-Ihog binding by competing for the Ihog Fn1 domain. Taken together, these results suggested Hh binding to Ihog is dominant over Ihog-Ihog homophilic interactions and effectively competes for Ihog even in the context of pre-established Ihog-Ihog *trans*-homophilic interactions.

## Discussion

We investigated the functional roles of the Hh receptor Ihog by determining a mechanism by which the Ihog proteins stabilizes cytonemes in the *Drosophila* wing imaginal disc. We found a dual role for cytoneme-localized Ihog proteins in Hh signal transduction and in *trans*-homophilic binding that mediates cytoneme-cytoneme interactions. This dual function is not unique to the Ihog proteins. Like Ihog, other members of the Ig-CAM family, such as the netrin receptor DCC, the Slit receptor Robo, and neural cell adhesion molecule N-CAM, also have dual roles in adhesion and signaling (*Juliano, 2002*; *Orian-Rousseau and Ponta, 2008*). The ligand binding and cell adhesion functions often involve different extracellular domains of the same protein, thus are biochemically separated and mutually compatible (*Frei et al., 1992*; *Martin-Bermudo and Brown, 1999*; *Sjöstrand et al., 2007*). However, the Ihog homophilic binding site overlaps with Ihog-Hh interface on Fn1 of the Ihog extracellular domain, resulting in competition between Hh binding and Ihog-Ihog *trans* homophilic interactions.

On the basis of these two functions competing for an overlapping surface on the Ihog Fn1 domain and the differential interaction strength between Ihog-Hh heterophilic and Ihog-Ihog *trans*-homophilic bindings, we propose a model in which Ihog-Ihog *trans* interactions promote and stabilize direct cytoneme-cytoneme contacts to facilitate Ihog in reaching and capturing the Hh ligands secreted from the cytonemes. The stronger ligand-receptor interaction releases Ihog from the weaker *trans*-homophilic interaction, enabling the receptor-ligand complex to freely transport or become internalized along the cytonemes. Meanwhile, the unbound former homophilic binding-partner Ihog either forms a new homophilic contact or engages in ligand-receptor complex formation along the cytonemes (*Figure 8*). Thus, the apparently incompatible functions of Ihog in homophilic adhesion and ligand binding cooperate to promote Hh transport and reception along the cytonemes. The model incorporates the role of the glypicans, Dlp, and Dally. Although contrary to a previous model that limited the contribution of the glypicans in cytoneme stabilization to a *trans* role (*González-Méndez et al., 2017*), in our model Dlp or Dally can contribute either as the membrane-tethered or the shed form and participate Ihog-Ihog *trans* interactions. Our model is also consistent with the reported affinity of Hh for the Ihog-Ptc receptor complex, which is higher than the affinity of Hh for the co-receptor Ihog alone (*Zheng et al., 2010*). Indeed, the presence of Ptc in the Hh-receiving cytoneme is critical to Hh reception in the responding cells (*Chen et al., 2017*). Thus, we propose that the integration of the functions of Ihog — promotion of cytoneme-cytoneme contacts, Hh delivery, and Hh signal reception — depends on the differential affinity (Ihog-Ihog < Ihog-Hh < Ptc-Ihog-Hh) and the competitive binding between Ihog for itself (in *trans*) and Hh (*Figure 8*).

The *trans*-homophilic Ihog-Ihog interaction is not only critical for cytoneme stabilization but also for A/P compartment boundary maintenance in the *Drosophila* wing discs (*Hsia et al., 2017*). Notably, both Hh release and cytoneme formation occurs at the basal side of the wing disc epithelium (*Callejo et al., 2011*; *Bilioni et al., 2013*; *Bischoff et al., 2013*; *Gradilla et al., 2014*; *Chen et al., 2017*; *González-Méndez et al., 2017*), making weaker Ihog-Ihog *trans* interactions accessible for replacement by stronger Ihog-Hh interactions along the cytonemes, where Ihog also functions as the receptor for Hh transport and reception (*Figure 8*). In contrast, farther from the source of secreted Hh, heterophilic Ihog-Hh interactions would be infrequent along the lateral side of epithelia, where the *trans* Ihog-Ihog interactions play essential roles in modulating A/P cell segregation and lineage restriction (*Hsia et al., 2017*). In addition, direct membrane contacts are much more extensive along the lateral sides of epithelia compared with that formed along cytonemes (*Figure 5*), favoring persistent Ihog-Ihog *trans* interactions that create an additional barrier for direct competition from the basally released Hh ligands. Thus, in agreement with the functional needs and the availability of Hh ligands, Ihog-Ihog homophilic *trans* interactions along the cytonemes are readily switchable, whereas those at the lateral cell-cell junctions are more stable and less likely to be disrupted (*Figure 8*).

Of note, Ihog-Ihog homophilic interactions could occur both in *trans* and in *cis* (*McLellan et al., 2006*; *Wu et al., 2019*). The Ihog-mediated homophilic *cis* interaction was observed in the HhN/IhogFn1–2 crystal structure (PDBID Code 2IBG), in which each HhN molecule contacts a single Ihog molecule, and a pair of 1:1 Hh/Ihog complexes form a dimeric 2:2 complex that is entirely mediated by *cis* interactions between the IhogFn1 domains. In this complex, Hh does not interfere with the *cis* Ihog-Ihog interaction (*McLellan et al., 2006*). In contrast, the Ihog-Ihog *trans* interaction was not revealed in the HhN/IhogFn1–2 crystal structure, likely due to competitive binding by HhN to the

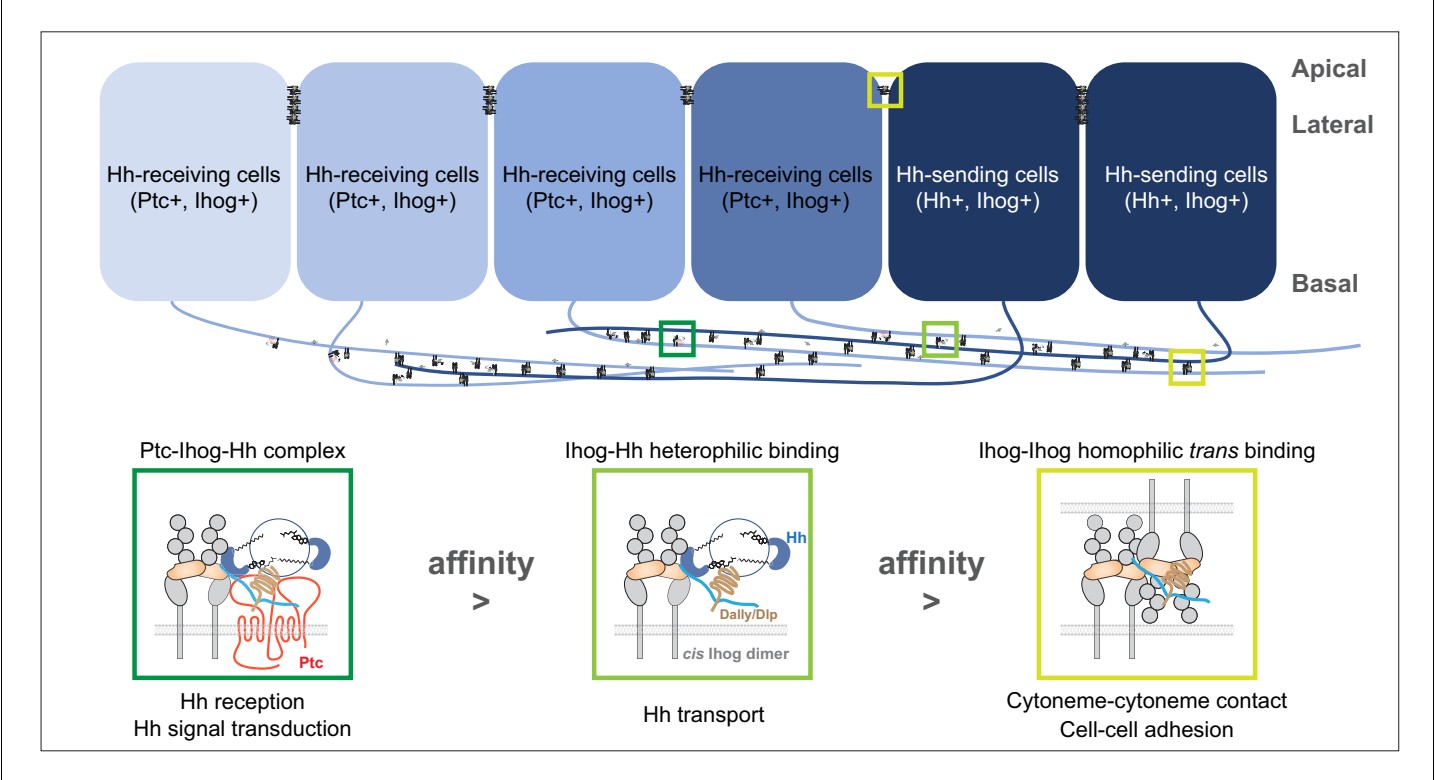

**Figure 8.** A model of the competitive coordination of the dual roles of Ihog in homophilic adhesion and signal reception. Diagram illustrating Ihog-Ihog homophilic *trans* interaction and Ihog-Hh heterophilic interaction in the wing imaginal disc epithelium. For simplicity, only a small number of the apical and lateral interactions are shown. Based on the differential affinity (Ptc-Ihog-Hh > Ihog-Hh > Ihog-Ihog) and the competitive binding between Ihog for itself (in *trans*) and for Hh, we propose a model in which Ihog-Ihog *trans* interactions promote and stabilize cytoneme-cytoneme contacts, thereby facilitating the 'capture' of Hh ligands, released from cytonemes of Hh-expressing cells, by Ihog on cytonemes of adjacent cells, ultimately reaching Ihog on cytonemes of Hh-receiving cells. The stronger Hh-Ihog interaction triggers release of Ihog from the weaker *trans*-homophilic interaction, enabling the receptor-ligand complex to transport along the cytoneme. Ultimately, the strongest interaction of Hh with the Ptc and Ihog complex results in Hh signal transduction. Both Hh release and cytoneme formation occur at the basal side of the wing disc epithelium, making weaker Ihog-Ihog *trans* interactions accessible for the replacement by stronger Ihog-Hh interactions along the cytonemes, where Ihog also functions as the receptor for Hh transport and reception. In contrast, farther from the source of secreted Hh, heterophilic Ihog-Hh interactions would be infrequent along the lateral side of epithelia, where the *trans* Ihog-Ihog interaction plays an essential role in maintaining A/P cell segregation and lineage restriction. The heparan sulfate necessary for the Ihog-Hh or *trans* Ihog-Ihog interactions may be supplied by Dally or Dlp, either as the membrane-associated forms of these glycans or as the form released upon shedding.

heparin-binding surface of IhogFn1 required to establish the *trans* Ihog-Ihog interaction. Although an Fn1-Fn1 interaction was noticed from the lattice contacts in the crystal structure of IhogFn1-2 by itself (PDBID Code 2IBB), this interaction is neither identical to the observed *cis* nor the predicted *trans* Ihog-Ihog binding, which may be because the IhogFn1-2 alone crystals were grown in the absence of heparin (*McLellan et al., 2006*). Nevertheless, we previously noticed, in an *in vitro* competitive binding assay, that HhN-mediated disruption of Ihog-Ihog binding was partial even when the concentration of HhN was 10–15 times higher than the reported dissociation constant of IhogFn1-2 for HhN (*Wu et al., 2019*), suggesting the co-existence of HhN-replaceable *trans* Ihog-Ihog binding and HhN-irreplaceable *cis* Ihog-Ihog binding. Thus, the *cis* Ihog-Ihog interaction likely can occur in the context of either *trans* Ihog-Ihog or Ihog-Hh interactions (*Figure 8*). Further studies are needed to investigate whether these two types of Ihog-Ihog interactions may influence each other, either positively or negatively, in the context of cell adhesion and cytoneme-cytoneme interactions.

Besides the wing imaginal disc epithelia, cytoneme-mediated Hh reception and transport have been described in other *Drosophila* tissues, such as the abdominal and the female germline stem cell niche (*Rojas-Ríos et al., 2012*; *Bischoff et al., 2013*; *González-Méndez et al., 2017*). The

involvement of cytonemes in Hh signaling has been extended from insect to vertebrates by studies of the limb bud of chick embryo and cultured mouse embryonic fibroblasts (*Sanders et al., 2013*; *Hall et al., 2020*). The vertebrate homologs of the *Drosophila* Ihog proteins, Cdo and Boc, localize in cytonemes, and overexpression of either CDO or BOC increases the number of cytonemes detected on mammalian cells (*Hall et al., 2020*). Further studies are necessary to explore whether the vertebrate Hh co-receptors CDO and BOC also have dual roles in adhesion and signaling along the cytonemes. Moreover, cytonemes have been implicated in the delivery of other paracrine signaling molecules important in development, including Notch, epidermal growth factor, fibroblast growth factor, bone morphogenetic protein, and Wnt (*González-Méndez et al., 2019*). Remarkably, cytonemes from a single cell often exhibit different receptor compositions such that different cytonemes from the same cell can selectively respond to ligands for a specific pathway and not others (*Roy et al., 2011*). The mechanism that segregates receptors to different cytonemes is yet not known. Whether homotypic adhesion contributes to the distinct localization of morphogen receptors to different cytoneme remains an open question awaiting further investigation.

# Materials and methods

**Key resources table**

| Reagent type (species) or resource | Designation | Source or reference | Identifiers | Additional information |
|---|---|---|---|---|
| Genetic reagent (*D. melanogaster*) | hs-FLP | *Golic and Lindquist, 1989* | FBti0000785 | Chr 1. |
| Genetic reagent (*D. melanogaster*) | Actin> y+> GAL4 | *Ito et al., 1997* | FBti0009983 | Chr 2. |
| Genetic reagent (*D. melanogaster*) | ptc-GAL4 | *Hinz et al., 1994* | FBal0287777 | Chr 2. |
| Genetic reagent (*D. melanogaster*) | CoinFLP-LexA:: GAD.GAL4 | BDSC | RRID:BDSC_58753 | Chr 2. |
| Genetic reagent (*D. melanogaster*) | tub-GAL80ts | BDSC | RRID:BDSC_7019 | Chr 2. |
| Genetic reagent (*D. melanogaster*) | UAS-GMA-GFP | BDSC | RRID:BDSC_31775 | Chr 2. |
| Genetic reagent (*D. melanogaster*) | UAS-mCD8-GFP | BDSC | RRID:BDSC_5137 | Chr 2. |
| Genetic reagent (*D. melanogaster*) | UAS-myr-mRFP | BDSC | RRID:BDSC_7119 | Chr 3. |
| Genetic reagent (*D. melanogaster*) | UAS-GPI-YFP | *Greco et al., 2001* | a gift from I. Guerrero | Chr 3. |
| Genetic reagent (*D. melanogaster*) | UAS-CD4-spGFP1-10 | BDSC | RRID:BDSC_58755 | Chr 3. |
| Genetic reagent (*D. melanogaster*) | UAS-Ihog WT | *Zheng et al., 2010* | | Chr 3. |
| Genetic reagent (*D. melanogaster*) | UAS-IhogΔFn1 | *Zheng et al., 2010* | | Chr 3. |
| Genetic reagent (*D. melanogaster*) | UAS-IhogΔFn2-HA | This paper | | ORF encoding IhogΔFn2 (delete aa 573–685) was cloned into pUAST to generate *UAS-IhogΔFn2*. |
| Genetic reagent (*D. melanogaster*) | UAS-Ihog-RFP | *Callejo et al., 2011* | a gift from I. Guerrero | Chr 3. |
| Genetic reagent (*D. melanogaster*) | UAS-Ihog$^{xHep}$ | This paper | | ORF encoding Ihog$^{Xhep}$ (R503E, K507E, K509E, and R547E) was cloned into pUAST to generate *UAS- Ihog$^{Xhep}$*. |

*Continued on next page*

*Continued*

| Reagent type (species) or resource | Designation | Source or reference | Identifiers | Additional information |
|---|---|---|---|---|
| Genetic reagent (*D. melanogaster*) | *UAS-Dally WT* | BDSC | RRID:BDSC_5379 | Chr 1. |
| Genetic reagent (*D. melanogaster*) | *UAS-Dlp WT* | BDSC | RRID:BDSC_9160 | Chr 3. |
| Genetic reagent (*D. melanogaster*) | *UAS-Ptc* | *Johnson et al., 1995* | | Chr 3. |
| Genetic reagent (*D. melanogaster*) | *UAS-Hh* | *Lee et al., 1992* | | Chr 3. |
| Genetic reagent (*D. melanogaster*) | *UAS-Ihog-RNAi* | VDRC | RRID:BDSC_29897 | Chr 3. |
| Genetic reagent (*D. melanogaster*) | *LexAop-CD4-spGFP11* | BDSC | RRID:BDSC_58755 | Chr 3. |
| Genetic reagent (*D. melanogaster*) | *LexAop. Ihog-RFP* | *González-Méndez et al., 2017* | a gift from I. Guerrero | Chr 3. |
| Genetic reagent (*D. melanogaster*) | *boi* | *Zheng et al., 2010* | | Chr 1. |
| Genetic reagent (*D. melanogaster*) | *dallyCPTI001039* | Kyoto Stock Center (DGRC) | RRID:BDSC_115064 | Chr 3. |
| Antibody | Mouse monoclonal anti-Dally-like (Dlp) | DSHB | Cat# 13G8; RRID:AB_528191 | IF: 1: 50 |
| Antibody | Mouse monoclonal anti-Ptc | DSHB | Cat# Apa 1; RRID:AB_528441 | IF: 1: 50 |
| Antibody | Rabbit polyclonal Anti-Hh | *Tabata and Kornberg, 1994* | a gift from T. Tabata | IF: 1: 500 |
| Antibody | Rabbit polyclonal anti-GFP | Molecular Probes | Cat# A-11122, RRID:AB_221569 | IF: 1: 2000 |
| Antibody | Rat polyclonal anti-Ihog | *Yao et al., 2006* | NA | IF: 1: 500 |
| Antibody | Mouse monoclonal anti-alpha Tubulin (DM1A) | Millipore | Cat# CP06, RRID:AB_2617116 | IF: 1:5000 |
| Antibody | Rabbit polyclonal anti-Hh | *Lee et al., 1992* | NA | WB: 1:1000 |
| Antibody | Mouse monoclonal anti-beta Tubulin | DSHB | Cat#E7; RRID:AB_2315513 | WB: 1:5000 |
| Antibody | Mouse monoclonal anti-HA.11 (16B12) | Covance | Cat# MMS-101P-1000, RRID:AB_291259 | IF: 1:1000 |
| Antibody | Fluorophore-conjugated secondary antibodies | Jackson Immuno-Research Lab | NA | IF:1: 500 |
| Antibody | HRP-conjugated secondary antibodies | Jackson Immuno-Research Lab | NA | WB: 1:10,000 |
| Antibody | Alexa Fluor 594 Phalloidin | Thermo Fisher Scientific | Cat# A12381, RRID:AB_2315633 | IF:1: 100 |
| Others | DAPI | Millipore Sigma | Cat# D9542 | |
| Other | Fetal Bovine Serum | Omega Scientific | Cat# FB-02 | |
| Other | Penicillin-Streptomycin-Glutamine (100X) | Thermo Fisher Scientific | Cat# 10378016 | |
| Other | Antifade mounting media | VECTASHIELD | Cat# H-1000 | |
| Other | FuGENE HD transfection reagent | Promega | Cat# E2311 | |
| Other | 16% Paraformaldehyde aqueous solution | Electron Microscopy Sciences | Cat# 15710 | |

*Continued on next page*

*Continued*

| Reagent type (species) or resource | Designation | Source or reference | Identifiers | Additional information |
|---|---|---|---|---|
| Recombinant DNA reagent | MBP-HhN expression plasmid | *McLellan et al., 2006* | a gift from D. Leahy | |
| Cell line (*D. melanogaster*) | *S2* | DGRC | Cat# S2-DGRC | |
| Software, algorithm | Fiji | NIH | RRID:SCR_002285 | |
| Software, algorithm | GraphPad Prism | GraphPad Software | RRID:SCR_002798 | |
| Software, algorithm | MATLAB | MATLAB Software | RRID:SCR_001622 | |

## Cell culture and transfection

*Drosophila* S2 cells (S2- DGRC) were obtained directly from the *Drosophila* Genomics Resource Center (DGRC) and regularly confirmed to be free of contamination (e.g. mycoplasma) through PCR-based tests as recommended by the NIH. The S2 cells were cultured in *Drosophila* Schneider's medium supplemented with 10% of fetal bovine serum (Omega Scientific) and 1% Penicillin-Streptomycin-Glutamine (Thermo Fisher) at 25℃ in a humidified incubator. Transfection was performed with FuGENE six transfection reagent (Promega). Expression constructs of GFP, mCherry, HhN, Hh, Ihog, and Ihog-YFP used in *Drosophila* cell culture were cloned into pAcSV plasmid as previously described (*Wu et al., 2019*).

## Cell aggregation assay

S2 cells were transfected separately with plasmids expressing desired proteins. 48 hr after transfection, S2 cells were washed with PBS and dissociated by 0.05% trypsin treatment for 5 min at 25℃. The dissociated cells were resuspended in fresh medium with 10% fetal bovine serum. The resuspended cells were then incubated in 1.5 ml ultra-low adhesion microcentrifuge tubes with gentle rotation at room temperature for the time indicated in the figure legends. Cells were then transferred into glass bottom dishes (D35-20-1.5-N, In Vitro Scientific) for live imaging by microscopy. In the experiments involving mixing differentially labeled red and green cells, cells co-expressing GFP or mCherry with the plasmid expressing the protein of interest were counted under microscope and mixed with equal number of transfected cells prior to incubation with rotation.

To assess cell aggregation, low-magnification fields of similar cell density were randomly taken from each cell aggregation experiment, and the cell clusters were scored if they contained three or more cells. The aggregation effect was quantified as the ratio of certain transfected cells within clusters to total transfected cells (both clustered and non-clustered). Each bar shows the mean ± SD from 20 to 30 different images. Unpaired two-tailed t test was used for statistical analysis. Statistical analysis was performed using GraphPad Prism software.

## Cell immunostaining and imaging

48 hr after transfection, dissociated S2 cells were allowed to settle and adhere for 60 min on a glass coverslip. Cells were then washed twice with PBS, fixed in 4% formaldehyde (Electron Microscopy Sciences) in PBS, blocked and permeabilized by 1.5% normal goat serum (NGS) and 0.1% Triton X-100 in PBS, incubated with primary antibody in PBS containing 1.5% NGS and 0.1% Triton X-100 for 1 hr at room temperature, washed 3 times with 0.1% Triton X-100/PBS, incubated with secondary antibody (with or without Phalloidin) and washed with 0.1% Triton X-100/PBS. The stained cells were mounted using the Vectashield anti-fade mounting medium (H-1000) and imaged with a Zeiss spinning disc confocal microscope.

## Computational modeling of cytonemes

We simulated the cytonemes using a *in silico* stochastic assay. The cell surface is simplified as a linear base line with length $M$ (shown as the black solid lines in the bottom of *Figure 5A*). In the initial step, we randomly picked $30\% \times M$ locations along the base line as the initial locations for the cytonemes. The initial cytoneme lengths are all 0.

In each simulation step, the binary interaction variable $S_k^{ij}$ was randomly assigned for each pair of neighboring segments according to *Equation 3*. From $S_k^{ij}$, the number of tip interactions for each cytoneme ($T_i$) and the total homophilic *trans* interaction for the $i^{th}$ cytoneme ($E_{interaction}^i$) can be obtained. For each cytoneme, its probabilities of elongation and shrinkage (i.e. $p_{elongation}$ and $p_{shrinkage}$) were computed using *Equations 1 and 2*; a random number $r \in [0, 1]$ was picked; if the cytoneme length was larger than 0 and $r \leq p_{shrinkage}/(p_{elongation} + p_{shrinkage})$, the cytoneme length was shrunk by 1 segment; otherwise it was elongated by 1. Once all cytonemes were elongated/shrunk, 1 cytoneme was randomly picked and tentatively moved along the cell surface base line; the interaction variables $S_k^{ij*}$ were re-assigned based this tentative configuration and the energy change $\Delta E_{translocation}^i$ was calculated; another random number $r \in [0, 1]$ was picked and the tentative move would be accepted if and only if $r \leq \min(1, \exp(-\Delta E_{translocation}^i/k_BT))$.

For each choice of $E_{ii}$, we allow the cytoneme system to evolve for more than 5,000,000 steps. Data were collected after 1,000,000 simulation steps when the cytoneme system reached steady state. In our showed results, we set $M = 100$, $p_{elongation}^0 = 5$, $p_{shrinage}^0 = 0.5$ and $\alpha = 0.2$. We tested different parameters and the general conclusions remained the same.

The cytoneme bundles were identified as a collection of parallel cytonemes located close together and forming more than three pairwise contacts between two cytonemes. The cytoneme bundling index is calculated by multiplying the minimum cytoneme length by the cytoneme number within each bundle.

## Computational modeling of cell rearrangement

We modeled the Ihog-Ihog and Ihog-Hh interactions using a vertex-based *in silico* assay (*Bi et al., 2015*; *Park et al., 2016*). Based on mechanical free energy calculated according to *Equation 5*, we used the Metropolis Monte Carlo method to perform the simulations. Simulations were performed within a $L \times L$ 2-D square space with a periodic boundary condition along the x- and y- axes. The initial conditions were sets of randomly generated morphologies: We first assigned $N$ cellular points and $5 \times N$ environmental points randomly distributed in the 2-D space, then used the Voronoi tessellation function in MatLab to partition the space into $6 \times N$ polygons based on these points, representing the $N$ cells and the empty space surrounding them.

We implemented Metropolis Monte Carlo simulations by moving the $6 \times N$ points according to the mechanical energy: In each tentative move, a random point is selected and a random displacement $\Delta l$ is assigned to it; the moved point causes re-partitioning of the 2-D monolayer using the Voronoi tessellation function, which results in a change in the monolayer's mechanical energy, $\Delta U'$; a random number $r \in [0, 1]$ is picked and the tentative move will be accepted if and only if $r \leq \min(1, \exp(-\Delta U'/k_BT))$.

We allowed the monolayer to randomly evolve for more than 650,000 steps. To ensure simulation efficiency, we adjusted the maximum allowed displacement to maintain the overall accept rate to be around $25 \sim 40\%$. Data were collected after 200,000 simulation steps when the monolayer's morphology had reached steady state.

In our simulations, $L$ is set to 20, and both the Hh-expressing S2 cells and the Ihog-expressing S2 cells are set with a unit area $A_0 = 1$. For simplicity, we assumed that areal elastic coefficient $\alpha = 500$ and the contractile coefficient $\beta = 6$ are the same for different types of cells. To account for the differential cell-cell adhesion, $\gamma_{HH}$ was set to 0 (Hh-expressing cells do not aggregate), $\gamma_{II} = 0.25$ for interaction between Ihog-expressing cells, we modulated the ratio $\gamma_{IH}:\gamma_{II}$ to set $\gamma_{IH}$ for our parameter study.

## *Drosophila* strains

The *ptc-GAL4* (*Hinz et al., 1994*) driver and tub-Gal80[ts] (*McGuire et al., 2003*) were used for transient expression of transgenic constructs using the GAL4/UAS system (*Brand and Perrimon, 1993*). Fly crosses were maintained at 18°C, and the *Gal80*[ts] repressor was inactivated for 24 hr at restrictive temperature (29°C) before dissection. The *actin>y+>GAL4* (*Ito et al., 1997*) driver was used to generate random ectopic clones of the *UAS* lines. The *CoinFLP-LexA::GAD.GAL4* driver was used to generate random clones expressing either *UAS-CD4-spGFP1-10* or *lexAop-CD4-spGFP11* (*Feinberg et al., 2008*; *Gordon and Scott, 2009*; *Bosch et al., 2015*). Larvae of the corresponding

genotypes were incubated at 37°C for 30–60 min to induce *hs-FLP*-mediated recombinant clones. The genotypes (see *Supplementary file 1*) of larvae for transient or random expression of transgenic constructs are listed in the Key Resources Table.

### Imaginal discs immunostaining and imaging

Wing discs from 3rd instar larvae were dissected, fixed in 4% formaldehyde (Electron Microscopy Sciences) in PBS, blocked and permeabilized by 5% normal goat serum (NGS) and 0.3% Triton X-100 in PBS, incubated with primary antibody in PBS containing 5% NGS and 0.3% Triton X-100 overnight at 4°C, washed three times with 0.3% Triton X-100/PBS, incubated with secondary antibody, and washed with 0.3% Triton X-100/PBS. The stained discs were mounted and imaged with a ZEISS spinning disc confocal microscope or a ZEISS LSM 880 with Airyscan. Average cytoneme length was determined using ImageJ and plotted using GraphPad Prism software.

### MBP-HhN purification

The MBP-HhN expression plasmid was a gift from Dr. Daniel Leahy (The University of Texas at Austin). A DNA fragment encoding the *Drosophila melanogaster* Hh residues 85–248 (HhN) was cloned into the MBP-HTSHP expression vector, which was modified based on the pMal-c2x vector (New England Biolabs) by including a linker region with various tags (His-TEV-Strep-His-PreScission). Similar to the procedure described previously (8), the fusion proteins were expressed in *Escherichia coli* strain B834 (DE3) by induction with 1 mM isopropyl 1-thio-β-D-galactopyranoside overnight at 16°C. Cells were harvested, lysed, and centrifuged, and the supernatant was passed over nickel-nitrilotriacetic acid resin (Qiagen). Proteins were eluted with imidazole according to the manufacturer's suggestions. The elution was then placed into 6000–8000 molecular weight–cutoff 40 mm dialysis tubing and dialyzed against 20 mM Tris (pH 8.0) and 200 mM NaCl.

### Western blot analysis

48 hr after transfection, S2 cells were lysed in 1% NP40 (50 mM Tris-HCl at pH 6.8, 150 mM NaCl, and protease inhibitors) for 30 min at room temperature. The lysate was clarified by centrifugation, and proteins were recovered directly in SDS-PAGE sample buffer. Proteins were separated by SDS-PAGE under reducing conditions and then transferred onto PVDF membranes (Millipore). After protein transfer, the membranes were blocked and then immunostained with primary antibodies and HRP-conjugated secondary antibodies.

### Statistical analysis

All data in column graphs are shown as mean values with SD and plotted using GraphPad Prism software. Statistical analyses were performed with unpaired two-tailed t-test, one-way ANOVA followed by Dunnett's, Sidak's or Tukey's multiple comparisons test, or two-sided Fisher's exact test was used for statistical analysis as described in the figure legends. The sample sizes were set based on the variability of each assay and are listed in the Figure legends. Independent experiments were performed as indicated to guarantee reproducibility of findings. Differences were considered statistically significant when $p < 0.01$.

## Acknowledgements

We thank S Blair, I Guerrero, D Leahy, T Tabata, the Bloomington *Drosophila* Stock Center, the Vienna *Drosophila* RNAi Center, Kyoto Stock Center and the Developmental Studies Hybridoma Bank for fly strains and reagents. We thank A Popratiloff from the GW Nanofabrication and Imaging Center, members of the Anatomy and Cell Biology Department for comments during the development of this work. We thank J S McLellan and N R Gough for helpful discussions.

## Additional information

### Funding

| Funder | Grant reference number | Author |
|---|---|---|
| National Institute of General Medical Sciences | R01GM117440 | Xiaoyan Zheng |

The funders had no role in study design, data collection and interpretation, or the decision to submit the work for publication.

### Author contributions

Shu Yang, Data curation, Formal analysis, Visualization, Writing - original draft; Ya Zhang, Data curation, Formal analysis, Visualization, Methodology; Chuxuan Yang, Data curation, Formal analysis, Methodology; Xuefeng Wu, Data curation, Visualization; Sarah Maria El Oud, Data curation; Rongfang Chen, Xufeng S Wu, Resources, Data curation, Software, Supervision, Visualization, Writing - review and editing; Xudong Cai, Resources, Data curation; Ganhui Lan, Conceptualization, Resources, Data curation, Software, Formal analysis, Supervision, Validation, Investigation, Methodology, Writing - original draft, Writing - review and editing; Xiaoyan Zheng, Conceptualization, Resources, Data curation, Software, Formal analysis, Supervision, Funding acquisition, Validation, Investigation, Visualization, Methodology, Writing - original draft, Project administration, Writing - review and editing

### Author ORCIDs

Shu Yang ⬛ https://orcid.org/0000-0001-8909-1962
Ya Zhang ⬛ https://orcid.org/0000-0001-9060-3777
Ganhui Lan ⬛ https://orcid.org/0000-0003-3768-4111
Xiaoyan Zheng ⬛ https://orcid.org/0000-0003-4983-5503

### Decision letter and Author response

Decision letter https://doi.org/10.7554/eLife.65770.sa1
Author response https://doi.org/10.7554/eLife.65770.sa2

## Additional files

### Supplementary files

• Supplementary file 1. Complete list of *Drosophila melanogaster* genotypes used in this study. The genotype of larvae from where wing discs were collected and imaged in each figure.

• Transparent reporting form

### Data availability

All data generated or analyzed during this study are included in the manuscript and supporting files.

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
