## [Decision Letter]

**Acceptance summary:**

This is a well-conceived and well-presented study that sheds light on two critical questions related to Hedgehog signaling, namely, the dual function of the Hedgehog co-receptor Ihog in Hh signal transduction and homophilic adhesion, and the regulation of cytoneme structure. Using experimental manipulation and mathematical modeling, the authors propose a model whereby the weaker Ihog-Ihog trans interaction promotes direct membrane contacts along cytonemes and that Hh-Ihog binding releases Ihog from trans Ihog-Ihog complex. The reviewers find the study convincing, interesting, and an important extension of biochemical/cellular work on Ihog to tissue behavior.

**Decision letter after peer review:**

Thank you for submitting your article "Competitive coordination of the dual roles of the Hedgehog co-receptor in homophilic adhesion and signal reception" for consideration by *eLife*. Your article has been reviewed by 3 peer reviewers, one of whom is a member of our Board of Reviewing Editors, and the evaluation has been overseen by Jonathan Cooper as the Senior Editor. The following individual involved in review of your submission has agreed to reveal their identity: Daniel J Leahy (Reviewer #3).

Essential Revisions:

All reviewers consider this a highly original and well executed study. No additional experiments are required, but the reviewers have raised a number of comments to improve presentation.

*Reviewer #1 (Recommendations for the authors):*

I have no major criticism of this study. It is appropriate for publication in *eLife*.

*Reviewer #3 (Recommendations for the authors):*

I have the following questions for the authors, some of which they may wish to address in the discussion:

1. In the crystal structure of the HhN/IhogFn12 complex an Fn1-mediated interaction between Ihog subunits that does not overlap with the HhN binding site was observed. Is it possible that this interaction is mediating the homophilic Ihog interaction and the competition for HhN binding is somehow mediated by heparin?

2. Are physiological levels of Ihog expression sufficient to promote bundling of cytonemes or do the large cytoneme bundles observed result from Ihog overexpression?

3. Is there a lattice contact in the structure of the IhogFn12 that may reflect the homophilic Ihog contact interface? As these crystal were grown in the absence of heparin, which is needed to form Ihog dimers, it seems unlikely but worth checking.

4. Are trans vs. cis Ihog interactions (i.e. interactions between Ihog protein on different vs. the same membrane) favored and if so how might that occur?

5. I am not a computational biologist but wonder if the modeling adds much to this paper. The conclusion seems to be that by making a few simple assumptions about interactions the model shows that the observed "bundling" effect is plausible. What is not clear to me is whether this effect is essentially predetermined by the assumption with the only variable being an arbitrary scaling constant.

---

## [Author Response]

Reviewer #3 (Recommendations for the authors):I have the following questions for the authors, some of which they may wish to address in the discussion:1. In the crystal structure of the HhN/IhogFn12 complex an Fn1-mediated interaction between Ihog subunits that does not overlap with the HhN binding site was observed. Is it possible that this interaction is mediating the homophilic Ihog interaction and the competition for HhN binding is somehow mediated by heparin?

The published HhN/IhogFn1–2 crystal structure reveals a 2:2 complex (McLellan et al. 2006), in which each HhN molecule contacts a single Ihog molecule, and a pair of 1:1 Hh/Ihog complexes form a dimeric 2:2 complex that is entirely mediated by cis interactions between the Ihog proteins. In this complex, Hh does not interfere with the cis Ihog-Ihog interaction.

In an in vitro competitive binding assay (Wu et al. 2019), we previously observed a concentration-dependent interference of purified HhN with interactions between differentially tagged IhogECD (soluble extracellular portion of Ihog). However, the HhN-mediated disruption of Ihog-Ihog binding was incomplete, even when the concentration of HhN was 10 times higher than the reported dissociation constant of IhogFn1-2 for HhN (Figure 7B, lane 2 in Wu et al., 2019). The Ihog-Ihog interaction that persisted in the presence of excess HhN was likely due to Ihog-Ihog homophilic cis interactions that were not competed by HhN.

Based on the observations listed above, we proposed a model that the trans- and cis- Ihog homophilic interactions can exist at the same time for a single Ihog molecule in vitro (Wu et al. 2019). However, it will be important to figure out whether these two types of interactions may influent each other, either positively or negatively, in the context of cell adhesion and cytoneme-cytoneme interactions in the future. We now revised Figure 8 and expanded the discussion on the trans and cis Ihog-Ihog interactions (P25-26).

“Of note, Ihog-Ihog homophilic interactions could occur both in trans and in cis (McLellan et al., 2006; Wu et al., 2019). […] Further studies are needed to investigate whether these two types of Ihog-Ihog interactions may influence each other, either positively or negatively, in the context of cell adhesion and cytoneme-cytoneme interactions.”

2. Are physiological levels of Ihog expression sufficient to promote bundling of cytonemes or do the large cytoneme bundles observed result from Ihog overexpression?

We thank the reviewer for pointing out that we were not sufficiently clear here. We have modified the related description on P16 (P17 of the revised version) as follows:

“…Therefore, the experimental observations are consistent with the computational modeling predictions that the augmented cytoneme-cytoneme interactions mediated by ectopic Ihog lead to bundled cytonemes. […] Therefore, endogenous Ihog proteins also played a crucial role in regulating the structure of cytonemes, which our data indicated involved Ihog-mediated cytoneme-cytoneme interactions.”

3. Is there a lattice contact in the structure of the IhogFn12 that may reflect the homophilic Ihog contact interface? As these crystal were grown in the absence of heparin, which is needed to form Ihog dimers, it seems unlikely but worth checking.

An Fn1-Fn1 interaction was noticed from the lattice contacts in the crystal structure of IhogFn1-2 by itself (PDB ID Code 2IBB), however, this interaction is neither identical to the observed cis nor the predicted trans Ihog-Ihog binding, maybe due to the fact that the IhogFn1-2 alone crystals were grown in the absence of heparin (McLellan et al., 2006). We now have included this information in the discussion. Please see the added Discussion section under response to #1.

4. Are trans vs. cis Ihog interactions (i.e. interactions between Ihog protein on different vs. the same membrane) favored and if so how might that occur?

We propose a model that both cis and trans Ihog-Ihog interactions can occur simultaneously for a single Ihog molecule. Please see the response to #1. We now revised Figure 8 and expanded the discussion on the trans and cis Ihog-Ihog interactions. Please see the added Discussion section under response to #1.

5. I am not a computational biologist but wonder if the modeling adds much to this paper. The conclusion seems to be that by making a few simple assumptions about interactions the model shows that the observed "bundling" effect is plausible. What is not clear to me is whether this effect is essentially predetermined by the assumption with the only variable being an arbitrary scaling constant.

We thank the reviewer for the opportunity to explain the importance of the modeling approach in this study.

First, the bundling effect was not predetermined in the model. In fact, the phenomenon of “cytoneme bundling,” which was neither previously reported nor observed in our own study (Figure 5A, F), was first revealed by the computational model. We then confirmed the phenomenon of “cytoneme bundling” by Airyscan (Figure 5G), which in turn helped to explain why Ihog-overexpressing cytonemes might be easier visualized and often used as a marker in studies of cytonemes.

Additionally, from the model development standpoint, it is critically important to limit the number of parameters in a model, not only because a model with too many parameters may be too flexible and overfit the results but also because concise models tend to have higher interpretability (i.e., a clearer linkage between the examined mechanism and the observations). Therefore, it is not uncommon in the biophysical modeling field that a model only has one or two parameters (oftentimes dimensionless parameters) as long as the parameters provide meaningful insights. In our model, the key parameter is a scaling constant reflecting the relative strength of the homophilic trans-interaction, which links Ihog-mediated trans homophilic binding to the increased cytoneme length/bundling caused by increased Ihog levels.

Due to the multiple binding models (heterophilic binding to Hh, homophilic cis binding, homophilic trans binding) and the overlapping binding surfaces of the Ihog proteins in Ihog-Hh and Ihog-Ihog trans binding, the influence of the Ihog homophilic trans interaction strength on cytoneme stabilization would not have been tested otherwise. We have modified related text on P16 (now P17), and hope the contribution of the modeling work could be better recognized. Please see the added section under response to #2.